# Modeling of streamflow in a 30-kilometer-long reach spanning 5 years using OpenFOAM 5.x

Yunxiang Chen, Jie Bao, Yilin Fang, William A. Perkins, Huiying Ren, Xuehang Song, Zhuoran Duan, Zhangshuan Hou, Xiaoliang He, and Timothy D. Scheibe

Pacific Northwest National Laboratory, Richland, Washington, USA 99354.

**Correspondence:** Yunxiang Chen (yunxiang.chen@pnnl.gov)

**Abstract.**

Developing accurate and efficient modeling techniques for streamflow at the tens-kilometer spatial scale and multi-year temporal scale is critical for evaluating and predicting the impact of climate- and human-induced discharge variations on river hydrodynamics. However, achieving such a goal is challenging because of limited surveys of streambed hydraulic roughness, uncertain boundary condition specifications, and high computational costs. We demonstrate that accurate and efficient three-dimensional (3D) hydrodynamic modeling of natural rivers at 30-kilometer and 5-year scales is feasible using the following three techniques within OpenFOAM, an open-source computational fluid dynamics platform: 1) generating a distributed hydraulic roughness field for the streambed by integrating water stage observation data, a rough wall theory, and a local roughness optimization and adjustment strategy; 2) prescribing the boundary condition for the inflow and outflow by integrating precomputed results of a one-dimensional (1D) hydraulic model with the 3D model; and 3) reducing computational time using multiple parallel runs constrained by 1D inflow and outflow boundary conditions. Streamflow modeling for a 30-kilometer-long reach in the Columbia River (CR) over 58 months can be achieved in less than six days using 1.1 million CPU hours. The mean error between the modeled and the observed water stages for our simulated CR reach ranges from -16 cm to 9 cm (equivalent to ca. $\pm 7\%$ relative to the average water depth) at seven locations during most of the years between 2011 and 2019. We can reproduce the velocity distribution measured by the acoustic Doppler current profiler (ADCP). The correlation coefficients of the depth-averaged velocity between the model and ADCP measurements are in the range between 0.71 and 0.83 at 75% of the survey cross-sections. With the validated model, we further show that the relative importance of dynamic pressure versus hydrostatic pressure varies with discharge variations and topography heterogeneity. Given the model's high accuracy and computational efficiency, the model framework provides a generic approach to evaluate and predict the impact of climate- and human-induced discharge variations on river hydrodynamics at tens kilometer and decade scales.

# 1 Introduction

As a major element of the water cycle, streamflow varies with upstream discharge, interacts with ambient physical and biological environments, and thus creates a variety of social, economic, and environmental functions (Wampler, 2012; Wohl et al., 2015; Harvey, 2016; Biddanda, 2017; Hiemstra et al., 2020). For instance, the flood control function is largely determined by accurate predictions of the water depth and flow speed that are further controlled by upstream discharge variations and the hydraulic roughness generated by flow-streambed interactions (USACE, 1994; Ferguson, 2019). The water quality management and biodiversity protection functions are strongly affected by the hydrological exchange flows (Harvey, 2016) that are driven by hydrostatic pressure and flow-sediment induced dynamic pressure (Tonina and Buffington, 2007; Cardenas and Wilson, 2007). As the magnitude, frequency, and peak time of discharge are projected to vary with future climate and anthropogenic conditions (Potter et al., 2004; Veldkamp et al., 2018; Wei et al., 2020; Xu et al., 2021), it is essential to establish a numerical modeling framework that enables evaluating and predicting the impact of climate- or human-induced discharge variations on streamflow and river functions.

Over the past three decades, computational fluid dynamics (CFD) models at various dimensions have been developed and applied to model streamflow (Bates et al., 2005). By solving the one-dimensional (1D) Saint-Venant equations, 1D numerical models have been widely used to predict flood routing (Richards, 1978; Keller and Florsheim, 1993; Carling and Wood, 1994; Hicks and Peacock, 2005), sediment transport (van Niekerk et al., 1992; Correia et al., 1992; Hoey and Ferguson, 1994; Ferguson et al., 2001; Talbot and Lapointe, 2002; Cui et al., 2003), water quality (Richmond et al., 2002), and aquatic habitats (Bovee, 1978; Milhous et al., 1984). A lot of software based on the 1D models, e.g., HEC-RAS, MIKE-11, ISIS, and InfoWorks, have also been developed and commercialized for practical applications. As the 1D models provide only cross-sectional averaged velocity and water depth, these models are usually problematic if flow manifests large variations in either the vertical or the cross-sectional direction (Lane and Ferguson, 2005). Due to these reasons, the two-dimensional (2D) numerical models, which solve the depth-averaged Navier-Stokes equations, have been developed to better capture the cross-sectional variations in flow (Miller, 1994; Bates et al., 1995; Lane and Richards, 1998; Thompson et al., 1998; Cao et al., 2003) and resulted influences on sediment transport (Alan D. Howard, 1992; Sun et al., 1996; Nagata et al., 2000; Duan et al., 2001; Darby et al., 2002), water quality (Perkins and Richmond, 2007), and aquatic habitats (Leclerc et al., 1995; Crowder and Diplas, 2000). Armed with increasingly powerful personal and high-performance computers, commercial 2D models such as HEC-RAS and SRH-2D are frequently deployed for flood management in urban and mountain areas. Despite the wide applications of 2D models, quasi-3D models are also gaining popularity because of increasing computer capacity and the capability to predict the vertical velocity component. Though quasi-3D models, e.g., Princeton Ocean Mode (Blumberg and Mellor, 1983), Environmental Fluid Dynamics Code–3D (Hamrick, 1992), Delft3D (Deltares, 2021), and CH3D (Johnson et al., 1993), have been commonly used for ocean, coastal, and river applications, they are not adequate to model the dynamic pressure.

As the dynamic pressure is a key driver of the flow, momentum, and nutrient exchange between stream water and ambient environments, e.g., meander river planform, complex instream structures, and groundwater, non-hydrostatic or fully 3D Navier-Stokes models are required in order to reliably predict river's environmental and ecological functions under dynamic discharge

conditions (Lorke and MacIntyre, 2009; Harvey, 2016; Hester et al., 2017; Chen et al., 2019). The full 3D simulations were firstly restricted to rivers with rectangular cross-sections (Leschziner and Rodi, 1979; Demuren and Rodi, 1986), and then were gradually extended for small-scale natural rivers with meander and roughness (Demuren, 1993; Olsen and Stokseth, 1995; Hodskinson, 1996; Hodskinson and Ferguson, 1998). A more realistic application is given by Sinha et al. (1998) whose work resolved the effects of large-scale roughness and multiple islands on streamflow in a 4-km stretch of the Columbia River. Later, more 3D models were applied to study hydrodynamics in natural streams (Nicholas and Sambrook Smith, 1999; Lane et al., 1999; Booker et al., 2001; Ma et al., 2002; Rodriguez et al., 2004; Huang et al., 2004; Lane and Ferguson, 2005; Lai, 2016), and its interactions with water quality (Hamrick, 1992; Ji et al., 2007; Sinha et al., 2013), vegetation flow (Wilson et al., 2006; Marjoribanks et al., 2017), fish habitat (Kolden et al., 2016), and hydrological exchange fluxes (Zhou et al., 2018; Bao et al., 2018, 2022). All 3D models mentioned above adopted the Reynolds-averaged Navier-Stokes (RANS) concept. More advanced models such as large-eddy-simulation (LES) have also been applied for natural streams by using high-performance computers and airborne Light Detection and Ranging (LiDAR) measured high-resolution topography (Khosronejad et al., 2016; Le et al., 2019; Khosronejad et al., 2020). The differences between RANS and LES in predicting stream velocities, turbulence, and secondary flows were also carefully examined using a field-scale experimental facility as a test bed (Kang et al., 2011; Kang and Sotiropoulos, 2011, 2012).

Though significant progress has been made in modeling streamflow, new challenges emerge as we apply existing CFD techniques to mitigate the impact of climate change and human activities on streamflow and river functions. Firstly, the modeling framework necessitates to efficiently model streamflow over large spatiotemporal scales because changes in hydrodynamics due to discharge variations often take months to decades to alter river bank structure, microbial community growth, fish life cycles, and eventually reshape river functions at grain to watershed scale (Wohl et al., 2005; Palmer et al., 2014; Wohl et al., 2015). Secondly, as applying the model at larger spatiotemporal scales usually means larger uncertainty from roughness calibration and inflow/outflow boundary condition specifications, it is necessary to develop an effective model data integration strategy such that the computational model can be better constrained by river bathymetry survey and water stage observations data. Additionally, applying the model to large spatiotemporal scales also requires a strategy to balance computational efficiency and model accuracy.

To address the above challenges, this work demonstrates a semi-automated workflow that enables accurate and efficient 3D CFD modeling of the streamflow in a 30-kilometer-long reach of the Columbia River spanning 9 years (Section 2). Specifically, a distributed hydraulic roughness calibration strategy is proposed to reduce the roughness calibration uncertainty by integrating water stage observations, a rough wall theory, and a local roughness optimization and adjustment procedure. An integrated 1D-3D model approach is also adopted to reduce the uncertainty from inflow/outflow boundary condition specifications and to provide boundary conditions for the temporal decomposition which targets computational efficiency improvement. The efficacy of the proposed workflow in calibrating roughness and predicting water stage and flow velocity during 2011–2019 is extensively demonstrated by comparing results from the present model and those from field observations in Section 3. Using the validated model, the relative importance of dynamic pressure to hydrostatic pressure and its dependency on discharge variations and topography heterogeneity are further investigated. The discussion on distributed roughness estimation, the model's medium

and long-term prediction performance, the relative importance of dynamic pressure, and the model's computational efficiency are given in Section 4.

## 2 Methods

### 2.1 River bathymetry, stage, and velocity surveys

The 30-kilometer-long reach is near the Hanford Site (www.hanford.gov) as shown (black box) in Figure 1a. The riverbed
bathymetry was measured using a Light Detection and Ranging (LiDAR) technique with less than 1 m resolution in vertical and 20 m resolution in horizontal directions. The measured bathymetry is then used as a geometric boundary in the CFD model. Water stage was measured in three periods at seven locations (red and yellow dots in Figure 1b) every 10 minutes. For convenience, observation 1 represents the measurements at 100B, 100N, 100D, Locke Island (LI), 100H, and 100F during 2011. Observation 2 denotes the measurement at 100B during 2013 and 2014. And those measured at 100HD during 2018 and
2019 are named as observation 3. These observations are then used for model calibration and validation. Specifically, water stages measured from January 20 to February 16, 2011 are used for model calibration. Measurements during the other dates in 2011 are used for short-term (less than 1 year after the calibration period) validation. Measurements during 2013 and 2014 are used for medium-term (2 to 3 years after the calibration period) validation. And those measured during 2018 and 2019 are used for long-term (7 to 8 years after the calibration period) validation. The survey at location 100HD is used to test the long-term
model performance in predicting water surface elevation (WSE) outside the calibration locations. Velocity distributions were also measured at 12 cross-sections (Figure 1c) along the river on March 4 (red lines) and April 1 (blue lines), 2011 using boat-towed acoustic Doppler current profiler (ADCP) for short-term velocity validation (Niehus et al., 2014). Horizontal coordinates and bed elevation of these locations are listed in Table A1. For convenience, the horizontal coordinate at the lower-left corner of the computational domain (Figure 1b blue box) is converted from (564,303.5598 m, 143,735.6771 m) in the geographic
information system map to (0,0) in the model domain. All vertical coordinates are referenced to the North American Vertical Datum of 1988.

### 2.2 Free surface tracking and turbulence model

Quantifying water surface elevation, velocity, and bed pressure requires an accurate solution to the water-air interface and turbulent flow. In this work, OpenFOAM-5.x (CFDDirect, 2017) is used to track the water-air interface using the volume of
fluid method (Hirt and Nichols, 1981; Deshpande et al., 2012) and simulate the turbulent flow using the time-averaged Navier-Stokes equations. The volume of fluid method marks a cell filled with liquid as $\alpha = 1$, filled with air with $\alpha = 0$, and partially filled liquid as $0 < \alpha < 1$. Denoting densities and viscosities of the liquid and gas by $\rho_l$, $\rho_g$, $\mu_l$, and $\mu_g$, then the density and viscosity of each cell is $\rho = \alpha\rho_l + (1-\alpha)\rho_g$ and $\mu = \alpha\mu_l + (1-\alpha)\mu_g$. Following these definitions, the time-averaged Navier-Stokes equations can be written as Eq. (1) and Eq. (2). The governing equation for volume fraction $\alpha$ can be written as Eq. (3).

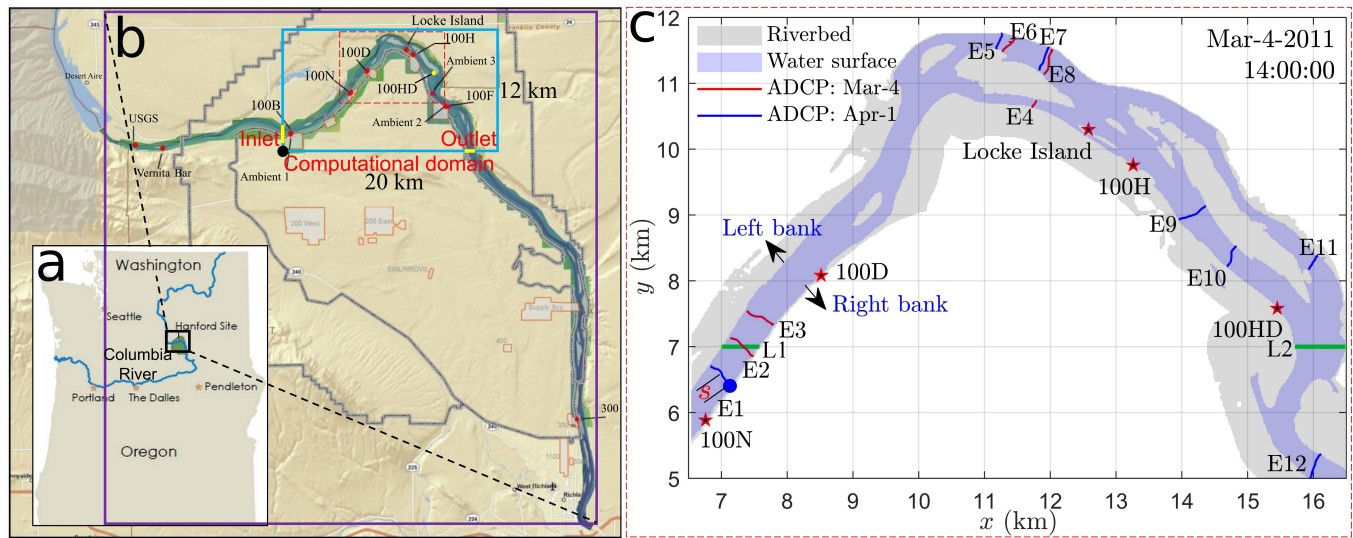

**Figure 1.** The location of the study site within Washington State and the Columbia River (a), the computational domain with the river bathymetry and water stage survey locations (b), and the exact locations of velocity and stage measurements (c). Yellow lines in (b) represent the inlet and outlet locations of the computational domain. Red and yellow dots in (b) denote water stage survey locations. Red and blues lines in (c) denote boat paths measured on two dates. Red stars and green lines in (c) represent the zoom-in of the stage survey locations in (b) and the locations (L1 and L2) selected to evaluate the sensitivity of stage and velocity variations to roughness heights, respectively. (a) is a reused image of the Oregon Department of Energy (www.oregon.gov/energy); (b) is modified from Figure 1 in Niehus et al. (2014) produced by S. Kallio at Pacific Northwest National Laboratory.


$$\nabla \cdot \boldsymbol{u} = 0 \tag{1}$$

$$\frac{\partial \rho \boldsymbol{u}}{\partial t} + \nabla \cdot (\rho \boldsymbol{u} \boldsymbol{u}) = \sigma \kappa_\alpha \nabla \alpha - \boldsymbol{g} \cdot \boldsymbol{x} \nabla \rho - \nabla p_d + \nabla \cdot \left[ (\mu + \mu_t) \nabla \boldsymbol{u} \right] - \nabla \cdot \left[ (\mu + \mu_t) (\nabla \boldsymbol{u}^T - \frac{2}{3} \nabla \cdot \boldsymbol{u} \mathbf{I}) \right] \tag{2}$$

$$\frac{\partial \alpha}{\partial t} + \nabla \cdot (\boldsymbol{u} \alpha) + \nabla \cdot \left[ \alpha (1 - \alpha) \boldsymbol{u}_r \right] = 0 \tag{3}$$

where $t$ is time, $\nabla = \frac{\partial}{\partial x} \boldsymbol{e}_x + \frac{\partial}{\partial y} \boldsymbol{e}_y + \frac{\partial}{\partial z} \boldsymbol{e}_z$ represents a spatial operator with $\boldsymbol{e}_x$, $\boldsymbol{e}_y$, and $\boldsymbol{e}_z$ denoting unit vectors along $x$, $y$, and $z$ directions. Also denoted are time average flow velocity ($\boldsymbol{u}$), surface tension coefficient ($\sigma$), interface curvature ($\kappa_\alpha$),

gravity acceleration ($\boldsymbol{g}$), spatial coordinate ($\boldsymbol{x}$), dynamic pressure ($p_d$), and dynamic turbulent viscosity ($\mu_t$). Specifically, the interface curvature is calculated by $\kappa_\alpha = -\nabla \cdot \left( \frac{\nabla \alpha}{|\nabla \alpha|} \right)$, the dynamic pressure $p_d$ is defined as $p_d = p - \rho \boldsymbol{g} \cdot \boldsymbol{x}$ with $p$ denoting the total pressure, $\boldsymbol{u}_r$ is the relative velocity of the liquid phase to the gas phase whose implementation in OpenFOAM can be found in Deshpande et al. (2012). The dynamic turbulent viscosity is determined by the $k - \omega$ shear stress transport (SST) model (Menter et al., 2003; Wilcox, 2006; CFDDirect, 2017).

## 2.3 Mesh generation and quality control


A good mesh quality is a crucial factor controlling computational stability and efficiency, especially for free surface tracking in large-scale river modeling over a long period (Deshpande et al., 2012). In this work, the mesh is generated using a two-step generation strategy, which first generates a structured background mesh and then removes all cells totally outside a given geometry (a river bathymetry in our case). Specifically, the background mesh is generated with a horizontal mesh resolution
of 20 m along $x$ and $y$. Such a resolution is identical to the horizontal resolution of the LiDAR-measured digital elevation model (DEM). The vertical mesh resolution is set as $\Delta z = 1$ m by balancing modeling accuracy and computational costs. One extra mesh resolution, 20 m$\times$ 20 m$\times$ 0.5 m, is also created to investigate the sensitivity of modeled riverbed pressure to mesh resolution (see uncertainty analyses in Appendix A1 and Figure A1). Figure 2 shows the horizontal and vertical mesh in the computational domain. It is observed that the aspect ratio for horizontal ($x$ and $y$) grid sizes is 1, but in the vertical direction,
it is 20. Figure 2c also shows that the zig-zag grid does not overlap with the riverbed, whose effect on flow is further discussed in the roughness calibration (see Section 2.4).

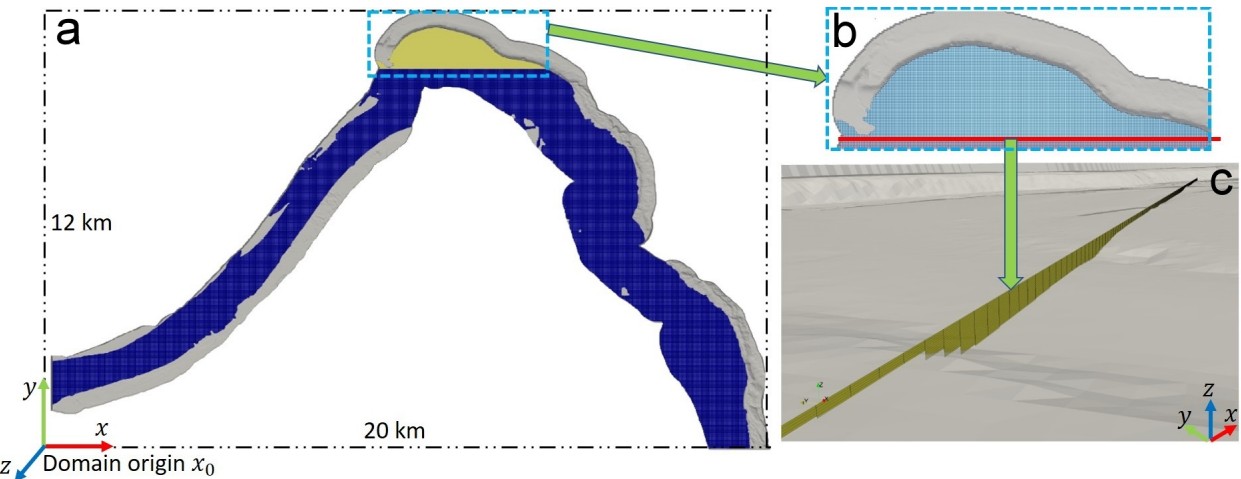

**Figure 2.** Horizontal and vertical computational meshes. (a) Top view showing the horizontal mesh over the whole domain. (b) Top view showing the details of horizontal mesh near LI. (c) 3D view showing details of the vertical mesh structure.

Though different from the traditional body-fitted mesh, such a zig-zag mesh strategy is both physically reasonable and technically necessary. Physically, the LiDAR-measured bathymetry cannot capture most geometric features that are smaller than 1 m, which means computational cells with a size less than 1 m are not necessary. In addition, the effect of geometric
features on flow dynamics, either from missing features less than 1 m or the differences attributed to mesh generation, has to be calibrated using the observed water stage through a distributed rough wall model (see Section 2.4). The efficacy of such a meshing and calibration approach in predicting water stage and velocity is demonstrated by comparing modeled water stage and velocity with field observations (see Sections 3.2–3.5). Technically, using grids with a high aspect ratio is usually necessary for river modeling. This is because the ratio of horizontal scales to the depth of rivers (around 1000 – 20000 in this work) is

usually large and a zig-zag mesh can maintain good mesh orthogonality at a large aspect ratio. By contrast, a body-fitted mesh with a large aspect ratio usually has bad mesh orthogonality, which causes code instability for free surface tracking and longer computational time.

## 2.4 Riverbed turbulence eddy viscosity and roughness parameterization

Rough elements are ubiquitous in natural rivers and have long been recognized as the major source of uncertainty in predicting
river discharge, flow speed, water surface profile, and sediment transport (USACE, 1994; Smith, 2014; Powell, 2014). In this work, the effect of rough elements on turbulent flow is quantified by linking riverbed turbulence eddy viscosity to bed roughness and flow conditions through a rough wall model (Versteeg and Malalasekera, 2007).

$$\nu_t = \nu[\frac{\kappa y_w^+}{\ln(Ey_w^+)} - 1] \tag{4}$$

Symbols in Eq. (4) denote turbulent kinematic viscosity $\nu_t = \mu_t/\rho$, kinematic viscosity $\nu = \mu/\rho$, von Karman's constant $\kappa = 0.41$, a non-dimensional wall distance $y_w^+ = \frac{y_w u_\tau}{\nu}$, and an integration constant $E$. Here $y_w$ and $u_\tau$ denote a wall distance and
riverbed shear velocity. The specific value of $E$ depends on the flow regime and the roughness parameter at the wall.

For natural rivers, the flow is usually in the fully rough turbulent flow regime. The integration value thus can be estimated by $E = E_0/(1 + C_s k_s^+)$ with $E_0$, $C_s$, and $k_s^+$ denoting a constant (with a value 9.8), a roughness distribution parameter, and a non-dimensional roughness height (Schlichting, 1979; Versteeg and Malalasekera, 2007; Blocken et al., 2007; CFDDirect, 2017). As classic theories on roughness are usually based on experiments of grain size roughness (Nikuradse, 1933), we choose
$C_s = 0.5$ with the assumption that natural roughness distribution is similar to uniformly roughed channels as in Nikuradse's experiments (Blocken et al., 2007). Therefore, the integration value mainly depends on $k_s^+$ which is defined as $k_s^+ = k_s u_\tau/\nu$. Here $k_s$ is the roughness height that needs to be calibrated with water stage observations.

As the bed shear velocity $u_\tau$ appears in both the non-dimensional wall distance $y_w^+$ and the non-dimensional roughness height $k_s^+$, estimation of the bed eddy viscosity shown in Eq. (4) is equivalent to estimating bed shear velocity and roughness
height. In this work, the bed shear velocity is estimated using the turbulent boundary layer theory that links a non-dimensional velocity ($u^+ = u/u_\tau$) to the non-dimensional wall distance ($y_w^+$) through a wall function $G$, i.e., $u^+ = G(y_w^+)$. In the fully rough regime, the wall function follows a logarithmic-law which has the form as $u^+ = \frac{1}{\kappa}\ln y_w^+ + B - \Delta B$ with $B = 5.2$ and $\Delta B = B - 8.5 + \frac{1}{\kappa}\ln k_s^+$ (Schlichting, 1979). Substituting the velocity ($u^0$) and wall distance ($y_w^0$) at the cell center closest to the wall, the wall function is converted to a non-linear function depending on shear velocity, roughness parameter, near-bed
velocity, and wall distance, as shown in Eq. (5). By solving such an equation under a given roughness $k_s$, we can obtain the value for bed shear velocity $u_\tau$ and wall turbulent eddy viscosity $\nu_t$.

$$G(u^0, y_w^0, u_\tau, k_s) = 0 \tag{5}$$

The above procedure means solving for shear velocity requires an estimation of bed roughness height $k_s$. For straight or short rivers, a uniform roughness height may be sufficient. However, for rivers with large curvature and complex cross-sectional shapes, e.g., islands, a distributed roughness height is necessary to capture the heterogeneous distribution of bed

shear velocity. This work proposes a generic approach to estimate a distributed roughness field using an error diagram and local roughness adjustment approach. The error diagram provides a rough estimation of the roughness parameters and the local adjustment further improves calibration accuracy per the error diagram. The error diagram is based on the fact that the water surface elevation increases with increasing roughness height and thus an optimal roughness height should fall in a range $0 < k_s < k_s^{max}$ in order for the model to match the observed water stage (Figure 3a and Figure A2).

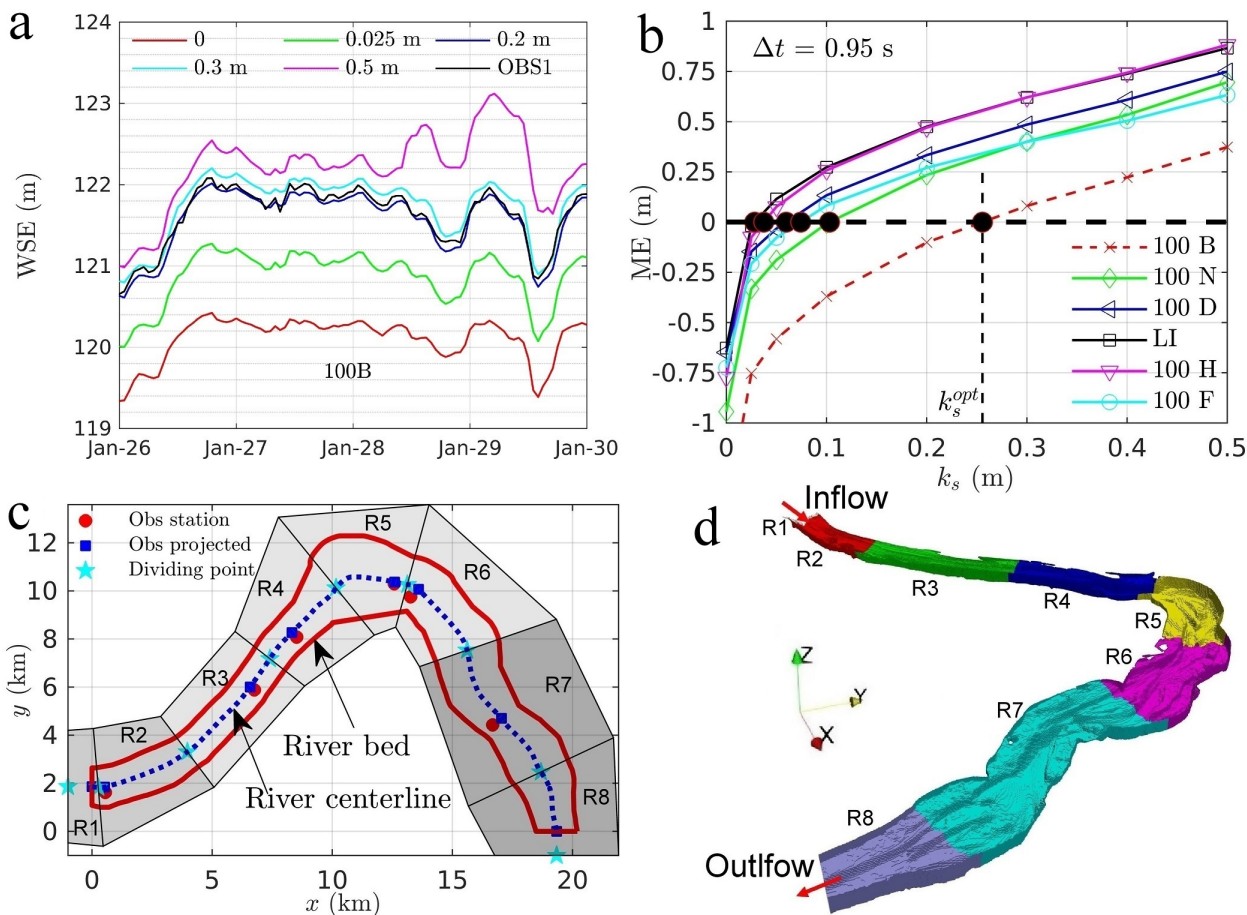

**Figure 3.** The effect of roughness height on WSE at a single location (a), ME between modeled and observed WSE at six locations (b), the procedure of generating eight roughness regions (c), and the 3D view of each region represented in mesh (d).

In this work, the effect of rough elements larger than 1 m in the vertical direction is directly resolved by mesh and thus an upper limit of roughness can be set as $k_s^{max} = 1$ m. With such an upper limit, we run our models at eight roughness values (0 m, 0.025 m, 0.05 m, 0.1 m, 0.2 m, 0.3 m, 0.4 m, 0.5 m) and then calculate the mean error (ME) and mean absolute error (MAE) between modeled water stage and observed ones at six locations (Figure 1b red dots) from January 20 to February 16, 2011. With the error diagram as shown in Figure 3b and Figure A3, we calculate an optimal roughness height $k_s$ for each observation location by making ME = 0 and MAE to be the minimum.

The optimal $k_s$ obtained in this way is then uniformly distributed in eight regions shown in Figure 3c. Here $k_s$ in R1 and R8 are identical to those in R2 and R7, respectively (Figure 3d). Due to the interactions of flow under different roughness parameters, the locally optimized roughness field does not guarantee low modeling errors at all locations (see case OF0 in Table 1). As higher deviations occur at 100B, 100N, and 100D, their roughness parameters are systematically adjusted to achieve better accuracy for all six locations (cases OF1–OF5 in Table 1). The final calibrated roughness values at the six calibration locations are listed in case OF in Table 1. These calibrated roughness parameters are then used to simulate the flow from May to December 2011, 2013–2015, and 2018–2019 to evaluate the modeling capability for short-term, medium-term, and long-term streamflow. A more comprehensive discussion of roughness estimation and local adjustment is included in Section 4.1.

## 2.5 Boundary conditions

Temporal variations in discharge at the inlet control the dynamic changes in streamflow and riverbed conditions. Figure 4 shows the temporal variations of discharge at the inlet during the years 2011 to 2019. A two-step approach is adopted to consider the discharge effects. Firstly, MASS1, a one-dimensional hydraulic model (Richmond and Perkins, 2009), is used to obtain the cross-sectional averaged velocity ($u^1$) and water stage ($z^1$) at 360 cross-sections along a 81 km long river section (Figure 1b green region) during 2011–2019. Then the velocity and stage are interpolated at the inlet and outlet locations (Figure 1b yellow lines) as $u_{in}^1$, $z_{in}^1$ and $u_{out}^1$, $z_{out}^1$, respectively. With these data, the inlet velocity and volume fraction are calculated as $\boldsymbol{u} = (u_x,0,0)$ with $u_x = u_{in}^1 \frac{\text{erf}[2(z_{in}^1-z)/\Delta z]+1}{2}$ and $\alpha = \frac{\text{erf}[2(z_{in}^1-z)/\Delta z]+1}{2}$. Here erf is an error function used to generate a sharp air-water interface. Other boundary conditions at the inlet are set as follows: uniform turbulence kinetic energy $k = 0.1\ \text{m}^2\text{s}^{-2}$, uniform specific dissipation rate $\omega = 0.003\ \text{s}^{-1}$; zero-gradient for dynamic pressure and turbulence eddy viscosity. It is worth mentioning that the given values of turbulent kinetic energy and specific dissipation rate have little effect on the results. At the outlet, velocity boundary condition is set as $\boldsymbol{u} = (0,-u_{out}^1,0)$ and all other boundaries are zero-gradient. At the top boundary (maximum elevation of the domain), pressure is set as 0 and the other variables are set as zero-gradient. At the riverbed, the turbulence eddy viscosity is determined through a rough wall model as discussed in Section 2.4. A no-slip boundary condition is set for velocity and zero-gradient boundary conditions are set for dynamic pressure, volume-fraction, and turbulence kinetic energy. The specific dissipation rate is calculated through $\omega_w = (\omega_{Vis}^2 + \omega_{Log}^2)^{1/2}$ with $\omega_{Vis} = \frac{6.0\nu}{\beta_1 y_w^2}$ and $\omega_{Log} = \frac{k^{1/2}}{C_\mu^{1/4}\kappa y_w}$ (see values of $\beta_1$ and $C_\mu$ in Table A2).

## 2.6 Spatiotemporal decomposition and initial conditions

Two spatiotemporal decomposition techniques are used in this work to improve computational efficiency. The first one is domain decomposition which decomposes the domain into 512 sub-domains and runs on 512 processors (see discussion on speedup in Section 4.4). Another one is time decomposition, which first divides the total simulation time, i.e., January 2013 to December 2015 and January 2018 to October 2019, into 58 months and then carries out parallel simulations for all 58 months simultaneously. The initial and boundary conditions for each month are set up at the time 4 days prior to the target simulation month. For example, to simulate the flow between February 1 and February 28, the simulation is extended to a period between January 28 and February 28, and initial and boundary conditions are set up at the start time on January 28.

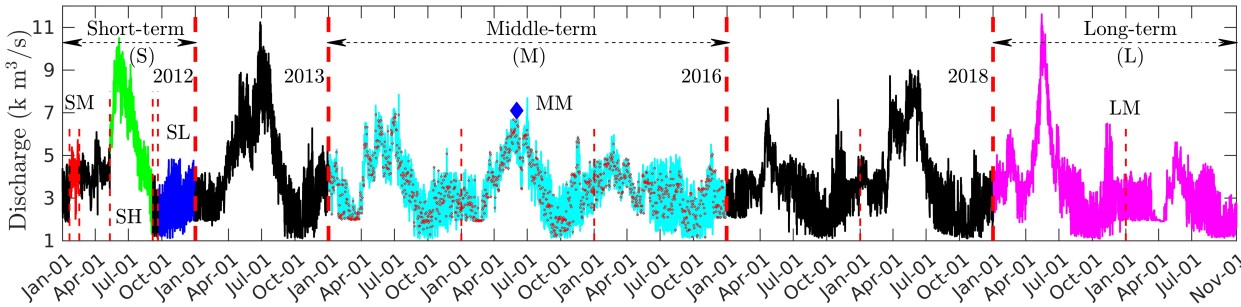

**Figure 4.** The time series of inlet flow rate during the years 2011–2019. S, M, and L denote short, medium, and long term. SM, SH, and SL denote the medium, high, and low flow in the short-term period; MM and LM denote mixed flow in the medium-term and long-term periods.

With such an approach, initial conditions for velocity, dynamic pressure, and eddy viscosity are set as zero, while for turbulence
kinetic energy and specific dissipation rate are 1e-20 $m^2s^{-2}$ and 0.003 $s^{-1}$ for all simulation months. The water stage and cross-sectional averaged velocity at the inlet and outlet boundaries at any time during the extended period are obtained from a one-dimensional hydraulic model as described in Section 2.5. It is important to note that such a spin-up approach works because (a) the flow reaches a quasi-steady state in $2 - 3$ flow-through times (about $T = L/U_0 = 30000/0.8$ s = 0.43 days); and (b) the time-varying boundary conditions at any time are available from existing data. Further discussion on the effect of
temporal decomposition on computational efficiency is included in Section 4.4.

## 2.7   Numerical schemes and solutions

The governing equations for flow ($u$, $p_d$), volume fraction ($\alpha$), and turbulence ($k$, $\omega$) were solved with an open-source CFD platform, OpenFOAM (Version 5.x), using a finite volume method (CFDDirect, 2017). The unsteady terms are discretized with a first-order Euler scheme, the advection term of flow is discretized with a second-order Gauss linear upwind scheme,
and the advection terms of turbulent kinetic energy and specific dissipation rate are discretized with a second-order Gauss linear scheme. The advection term and the compression term of volume fraction are discretized with Gauss vanLeer and Gauss linear schemes, respectively. All diffusion terms are discretized with a corrected central differencing scheme and all gradient terms are discretized with a second-order central differencing method. With these discretization schemes, initial, and boundary conditions, OpenFOAM first updates the volume fraction at the interface using a Multidimensional Universal Limiter with
Explicit Solution (MULES) algorithm (Zalesak, 1979; Kuzmin et al., 2003; Liu et al., 2016), and then solves the velocity-pressure coupling using a Pressure Implicit with Splitting of Operators (PISO) algorithm (Issa, 1985), followed by solving $\omega$ and $k$ equations. At each iteration, the discretized linear equation group for pressure is solved using a Diagonal-based Incomplete Cholesky Preconditioned conjugate gradient (DIC-PCG) method with a relative convergence tolerance of $10^{-10}$, and the discretized linear equation groups for velocity, volume fraction, turbulent kinetic energy, and specific dissipation rate
are solved with a symmetric Gauss-Seidel smooth solver at a relative tolerance $10^{-10}$. The initial time step is set as $10^{-10}$ s but allowed to adjust during runtime to not exceed 3 s. The maximum and average Courant number for all cases are less than

1.1 and 0.019, respectively. Here the Courant number is calculated as $C_o = \Delta_t \sum_f |\phi_i|/V$ with $\Delta_t$, $\sum_f |\phi_i|$, and $V$ denoting the variable time step, the total fluxes of all faces, and cell volume, respectively. With the solution of volume fraction, the water surface elevation is calculated by setting $\alpha = 0.5$ (Hirt and Nichols, 1981). It is necessary to note that the modeled water surface elevation changes little at time steps 0.1 s, 0.5 s, 0.95 s, 2 s, and 3 s (see Figure A4); therefore, the maximum time step is chosen as 3 s to reduce computational costs.

## 3    Results

### 3.1    Short-term roughness calibration

The error diagram approach gives a rough estimation of the hydraulic roughness at each location. The modeling accuracy using these roughness parameters are -16.5 cm – 6.4 cm and 7.6 cm – 19.6 cm at six locations (Case OF0 in Table 1) in terms of ME and MAE, respectively. By systematically adjusting the roughness parameters at 100B, 100N, and 100D, the overall modeling accuracy is improved. Figure 5a compares the water surface elevation using the locally adjusted roughness field (Case OF in Table 1) and those from observation 1. The comparison of the hourly recorded water stage data shows the modeled WSE accurately predicts the magnitude and frequency in the WSE. The 1:1 plot (Figure 5b) shows there is no systematic bias in the model, which can be further demonstrated by an R-squared ($R^2$) and linear-regression slope very close to 1 (Table 2 SM cases). Here $R^2 = 1 - \frac{\sum(WSE_m - WSE_o)^2}{\sum(WSE_m - \overline{WSE}_o)^2}$, $\overline{WSE}_o = \frac{\sum WSE_o}{N_t}$ with $WSE_m$, $WSE_o$, and $N_t$ denoting modeled WSE, observed WSE, and the number of time series, respectively. Quantitatively, the ME at the six locations falls in the range -7.5 cm – 6.4 cm, which is equivalent to -2.7% – 2.1% relative to the average water depth at each location (Table 2 SM cases RME). The MAE at all locations is 7.5 cm – 12.7 cm, which is equivalent to 2.1% – 5.3% relative to water depth (Table 2 SM cases RMAE). The root mean square, defined as RMS = $\sqrt{\frac{\sum(WSE_m - WSE_o)^2}{N_t}}$, for all locations is 9.2 cm – 16.4 cm, which is equivalent to 2.8% – 6.3% relative to the average water depth at each location (Table 2 SM cases RRMS). The comparisons of the simulated water depth with observed ones are shown in Figure 5(c,d), which demonstrates similar visual and quantitative accuracy as those observed in Figure 5(a,b).

### 3.2    Short-term water stage validation

Though this work calibrates the distributed roughness field using the observed WSE at a medium flow (discharge 4227 m$^3$/s) scenario, we show that calibrated roughness works well for predicting the WSE at high flow (6335 m$^3$/s) and low flow (2613 m$^3$/s) scenarios. Figure 6 compares the hourly recorded WSE with observations during high flow (Figure 6a) and low flow (Figure 6c). Figure 6b,d shows the 1:1 comparison between these data. The results show a good match in terms of the magnitude and frequency of the WSE at the six locations. The 1:1 plot shows there is no obvious bias in modeled WSE. In statistics, the ME during high flow is -2.5 cm – 9.1 cm, which is equivalent to -0.6% – 1.9% relative to mean water depth at each location. Similarly, these values at low flow is -15.6 cm – 5.5 cm and -7.1% – 6.6%, respectively. In terms of the MAE, it is 7.2 cm – 13.5 cm (1.5% – 3.1% relative to average water depth) at high flow and 13.1 cm – 26.6 cm (5.1% – 15.8% relative to water

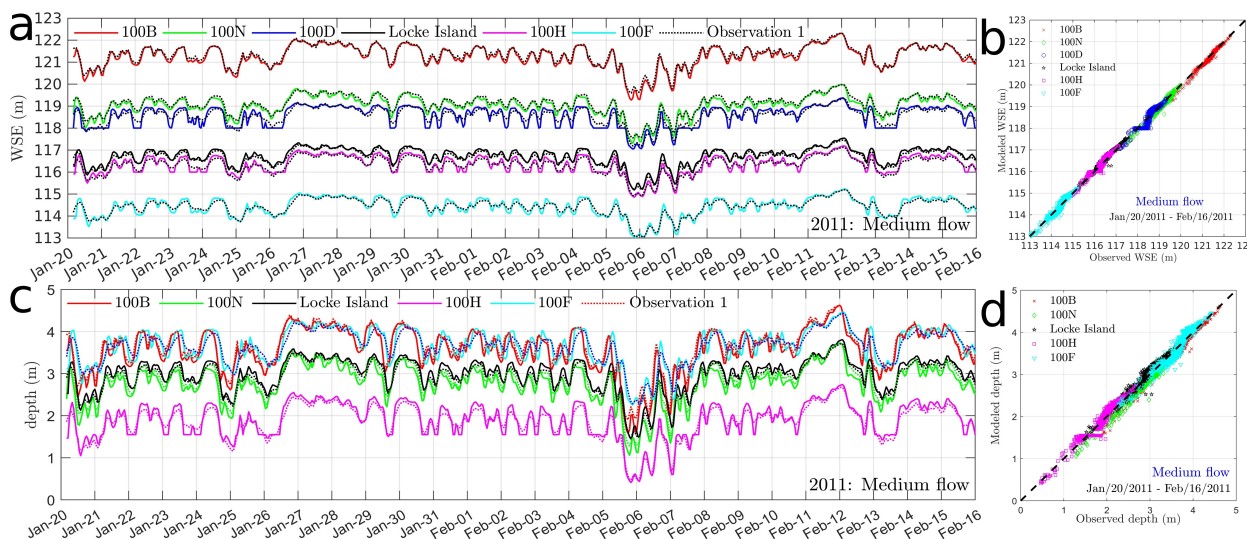

**Figure 5.** The comparisons of water surface elevation and depth between the model and observations using the calibrated roughness field (case OF in Table 1) during a medium flow in 2011. a and c represent the hourly recorded WSE and depth, respectively. b and d denote their 1:1 plots.

**Table 1.** Roughness adjustment approach and associated ME and MAE.

| Case | Calibrated $k_s$ (cm) | | | | | | ME during 1/20–2/16, 2011 (cm) | | | | | | | MAE during 1/20–2/16, 2011 (cm) | | | | | | |
|---|---|---|---|---|---|---|---|---|---|---|---|---|---|---|---|---|---|---|---|---|
| name | 100B | 100N | 100D | LI | 100H | 100F | 100B | 100N | 100D | LI | 100H | 100F | Range | 100B | 100N | 100D | LI | 100H | 100F | Range |
| OF0 | 25.56 | 10.3 | 5.98 | 2.83 | 3.74 | 7.42 | -16.5 | -19.5 | -6.8 | 6.4 | 4.8 | 0.3 | -16.5 – 6.4 | 16.5 | 19.6 | 12.7 | 7.6 | 8.9 | 9.3 | 7.6 – 19.6 |
| OF1 | 30 | 10.3 | 5.98 | 2.83 | 3.74 | 7.42 | -11.1 | -19.5 | -7.4 | 6.4 | 3.3 | 0.3 | -19.5 – 6.4 | 11.2 | 19.6 | 13.4 | 7.6 | 10.1 | 9.3 | 7.6 – 19.6 |
| OF2 | 40 | 10.3 | 5.98 | 2.83 | 3.74 | 7.42 | -0.6 | -19.5 | -7.4 | 6.4 | 3.3 | 0.3 | -19.5 – 6.4 | 5.8 | 19.6 | 13.4 | 7.6 | 10.1 | 9.3 | 7.6 – 19.6 |
| OF3 | 40 | 18.6 | 5.98 | 2.83 | 3.74 | 7.42 | 1.6 | -16.8 | -7.2 | 6.4 | 3.4 | 0.4 | -16.8 – 6.4 | 6.5 | 17.0 | 13.3 | 7.6 | 10.1 | 9.3 | 7.6 – 17.0 |
| OF4 | 30 | 18.6 | 5.98 | 2.83 | 3.74 | 7.42 | -8.7 | -16.9 | -7.4 | 6.4 | 3.3 | 0.3 | -16.9 – 6.4 | 9.1 | 17.0 | 13.3 | 7.5 | 10.1 | 9.2 | 7.5 – 17.0 |
| OF5 | 30 | 18.6 | 9.0 | 2.83 | 3.74 | 7.42 | -7.5 | -11.7 | -3.6 | 6.4 | 3.3 | 0.3 | -11.7 – 6.4 | 8.2 | 12.2 | 12.6 | 7.5 | 10.0 | 9.2 | 7.5 – 12.6 |
| OF | 30 | 18.6 | 12.0 | 2.83 | 3.74 | 7.42 | -6.6 | -7.5 | -0.6 | 6.4 | 3.3 | 0.3 | -7.5 – 6.4 | 7.7 | 8.9 | 12.7 | 7.5 | 10.0 | 9.2 | 7.5 – 12.7 |
| OFK1 | 12.2 | 12.2 | 12.2 | 12.2 | 12.2 | 12.2 | -29.1 | 6.6 | 17.7 | 32.1 | 31 | 13.1 | -29.1 – 32.1 | 29.1 | 7.5 | 19.5 | 32.6 | 31.5 | 14.6 | 7.5 – 29.1 |
| OFK2 | 25.56 | 6.25 | 6.25 | 6.25 | 6.25 | 6.25 | -16.7 | -15.0 | 1.3 | 14.8 | 12.2 | 13.1 | -16.7 – 14.8 | 16.7 | 15.2 | 12.2 | 15.1 | 13.4 | 9.2 | 9.2 – 15.2 |
| OFK50 | | | see Figure A8 | | | | -19.4 | -18.8 | -6.1 | 8.5 | 8 | 1.4 | -19.4 – 8.5 | 19.4 | 18.8 | 12.5 | 9.3 | 10.5 | 9.7 | 9.3 – 19.4 |
| MS | 30.5 | 18.6 | 15.6 | 3.9 | 3.9 | 7.42 | -4.7 | -1.2 | 4.9 | 7.7 | 3.9 | 0.3 | -4.7 – 7.7 | 6.7 | 6.4 | 13.9 | 8.6 | 10.3 | 9.2 | 6.4 – 13.9 |
| MS2 | 30.5 | 18.6 | 12.0 | 3.9 | 3.9 | 7.42 | -5.6 | -5.6 | 1.9 | 7.7 | 3.8 | 0.2 | -5.6 – 7.7 | 7.1 | 7.8 | 12.9 | 8.5 | 10.2 | 9.2 | 7.1 – 12.9 |
| MS3 | 30.5 | 18.6 | 9.0 | 3.9 | 3.9 | 7.42 | -6.6 | -9.8 | -1.0 | 7.7 | 3.8 | 0.2 | -9.8 – 7.7 | 7.6 | 10.6 | 12.5 | 8.6 | 10.2 | 9.2 | 7.6 – 12.5 |

depth) at low flow. The RMS is 9.7 cm – 15.9 cm (2.0% – 3.8% relative to water depth) at high flow and 17.7 cm – 40.3 cm (6.9% – 22.2% relative to water depth) at low flow. The calculated $R^2$ between the modeled and observed WSE is larger than

280   0.98 for six locations at high flow and is in the range 0.88 - 0.93 at low flow, except for at 100D where the value is 0.603. The

slope of the linear regression has a similar trend as $R^2$ that it falls in the range 1.05 – 1.1 during high and low flow at most locations, however, has a value of 0.859 at 100D during low flow. These results suggest that the modeled WSE agrees with observation very well at all locations during the high flow event. The model WSE is less accurate at low flow and has obvious deviation at locations where the water depth is less than 1 m (case SL at 100H) or not available due to being too close to the wet/dry boundary (100D).

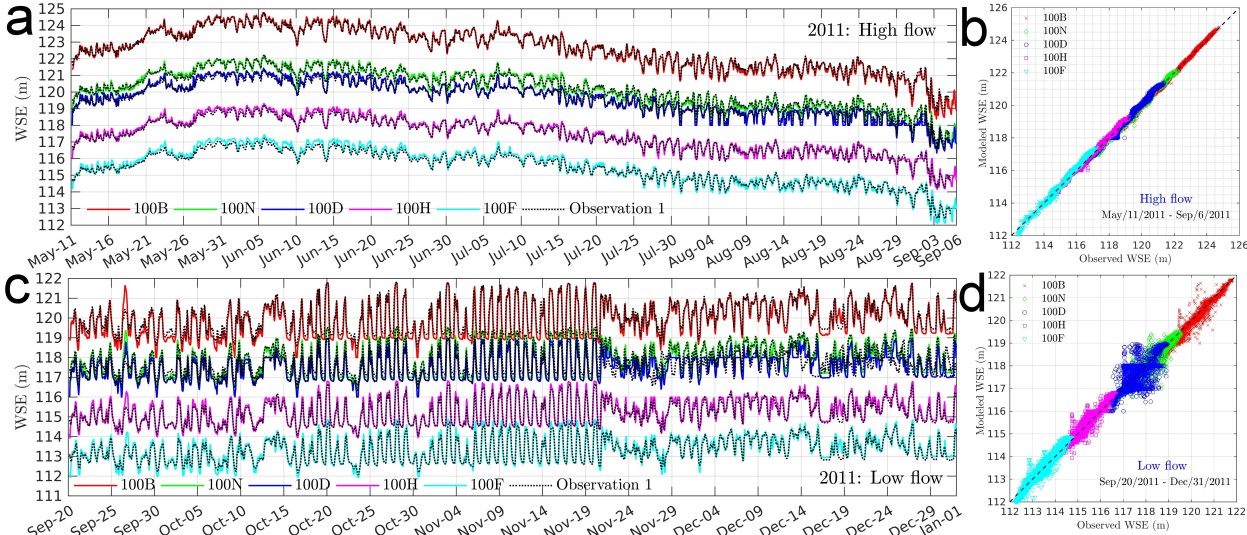

**Figure 6.** The comparison of water surface elevation from model and observations during high flow and low flow in 2011. (a-b) Hourly recorded time series of WSE and 1:1 plot during high flow. (c-d) Hourly recorded time series of WSE and 1:1 plot during low flow.

### 3.3    Short-term velocity validation

To further examine the model's predictive capability for flow velocity, Figure 7 shows a qualitative comparison of the velocity magnitude ($U$) distribution at 12 cross-sections between ADCP measurements and the CFD model. For instance, at the cross-section E1, ADCP data is placed at the left hand (a) and the corresponding CFD data is placed at the right hand (b). The distributions of velocity magnitude at other locations are arranged similarly. By comparing each pair of figures, it is found that the pattern of the distribution, e.g., locations of maximum and minimum velocity, is very similar. This means the CFD model can qualitatively reproduce the velocity distribution at each cross-section. In addition, it is observed that the distribution is "cleaner" in CFD data (e.g., Figure 7x), but shows more noise in ADCP measurements (e.g., Figure 7w). Such a noise feature is likely induced by small-scale turbulence, measurement uncertainty from boat movement (Khosronejad et al., 2016; Le et al., 2019), and other factors such as wind shear, riparian vegetation, and inaccuracy of topography survey (Lane et al., 1999). The ADCP measurement uncertainty can also be manifested by the white space on each figure where data are lost.

Due to these problems, a more commonly used way is to compare the depth average flow velocity from ADCP and CFD models as shown in Figure 8. The result shows that the agreement between ADCP and the simulation is very good at locations

**Table 2.** A summary of flow scenario, discharge, water depth, roughness height, and modeling accuracy for calibration, validation, and prediction.

| Survey Station | Time Period | Year | Month Day | Flow Scenario | Mean Discharge (m³/s) | Mean depth (m) | $k_s$ cm | WSE: OF-Observed ME cm | RME % | MAE cm | RMAE % | RMS cm | RRMS % | $R^2$ | $\beta$ |
|---|---|---|---|---|---|---|---|---|---|---|---|---|---|---|---|
| | SM | 2011 | 1/20–2/16 | Medium | 4227 | 3.57 | | -6.6 | -1.8 | 7.7 | 2.1 | 10.1 | 2.8 | 0.963 | 1.072 |
| | SH | 2011 | 5/11–9/6 | High | 6335 | 4.88 | | -2.1 | -0.4 | 7.2 | 1.5 | 9.7 | 2.0 | 0.994 | 1.062 |
| | SL | 2011 | 9/20–12/31 | Low | 2613 | 2.19 | | -15.6 | -7.1 | 19.7 | 9.0 | 25.4 | 11.6 | 0.914 | 1.102 |
| 100B | MH[2] | 2013 | 3/11–6/19 | High | 4449 | 3.65 | 30 | -10.1 | -2.8 | 11.9 | 3.3 | 15.1 | 4.1 | 0.982 | 1.083 |
| | ML[2] | 2013–14 | 9/27–1/5 | Low | 2517 | 2.10 | | -20.7 | -9.9 | 22.4 | 10.7 | 26.4 | 12.6 | 0.879 | 1.108 |
| | MH[2] | 2014 | 4/15–7/24 | High | 5217 | 4.27 | | -9.2 | -2.2 | 10.5 | 2.5 | 13.3 | 3.1 | 0.945 | 1.053 |
| | MM[2] | 2013–14 | 1/1–8/1 | Mixed | 3755 | 3.12 | | -14.4 | -4.6 | 16.1 | 5.2 | 22.1 | 7.1 | 0.965 | 1.065 |
| | SM | 2011 | 1/20–2/16 | Medium | 4227 | 2.78 | | -7.5 | -2.7 | 8.9 | 3.2 | 10.9 | 3.9 | 0.943 | 1.031 |
| 100N | SH | 2011 | 5/11–/9/6 | High | 6335 | 3.96 | 18.6 | -2.5 | -0.6 | 8.9 | 2.2 | 11.0 | 2.8 | 0.991 | 1.058 |
| | SL | 2011 | 9/20–12/31 | Low | 2613 | 1.58 | | -10.3 | -6.5 | 19.9 | 12.6 | 26.1 | 16.4 | 0.881 | 1.061 |
| | MM | 2013–15 | 1/1–12/31 | Mixed | 3424 | 2.17 | | NA | NA | NA | NA | NA | NA | NA | NA |
| | SM | 2011 | 1/20–2/16 | Medium | 4227 | NA | | -0.6 | NA | 12.7 | NA | 16.4 | NA | 0.874 | 1.149 |
| 100D | SH | 2011 | 5/11–/9/6 | High | 6335 | NA | 12 | 3.1 | NA | 10.6 | NA | 13.7 | NA | 0.983 | 1.071 |
| | SL | 2011 | 9/20–12/31 | Low | 2613 | NA | | -1.8 | NA | 26.6 | NA | 40.3 | NA | 0.603 | 0.859 |
| | MM | 2013–15 | 1/1–12/31 | Mixed | 3424 | NA | | NA | NA | NA | NA | NA | NA | NA | NA |
| | SM | 2011 | 1/20–2/16 | Medium | 4227 | 2.99 | | 6.4 | 2.1 | 7.5 | 2.5 | 9.2 | 3.1 | 0.948 | 1.023 |
| LI | SH | 2011 | 5/11–/9/6 | High | 6335 | 4.02 | 2.83 | NA | NA | NA | NA | NA | NA | NA | NA |
| | SL | 2011 | 9/20–12/31 | Low | 2613 | 1.91 | | NA | NA | NA | NA | NA | NA | NA | NA |
| | MM | 2013–15 | 1/1–12/31 | Mixed | 3424 | 2.44 | | NA | NA | NA | NA | NA | NA | NA | NA |
| | SM | 2011 | 1/20–2/16 | Medium | 4227 | 1.90 | | 3.3 | 1.7 | 10.0 | 5.3 | 12.0 | 6.3 | 0.923 | 1.073 |
| 100H | SH | 2011 | 5/11–/9/6 | High | 6335 | 3.00 | 3.74 | 5.7 | 1.9 | 9.2 | 3.1 | 11.3 | 3.8 | 0.989 | 1.053 |
| | SL | 2011 | 9/20–12/31 | Low | 2613 | 0.83 | | 5.5 | 6.6 | 13.1 | 15.8 | 18.4 | 22.2 | 0.922 | 1.062 |
| | MM | 2013–15 | 1/1–12/31 | Mixed | 3424 | 1.36 | | NA | NA | NA | NA | NA | NA | NA | NA |
| | SM | 2011 | 1/20–2/16 | Medium | 4227 | 3.66 | | 0.3 | 0.08 | 9.2 | 2.5 | 11.5 | 3.1 | 0.928 | 1.106 |
| 100F | SH | 2011 | 5/11–/9/6 | High | 6335 | 4.77 | 7.42 | 9.1 | 1.9 | 13.5 | 2.8 | 15.9 | 3.3 | 0.978 | 1.103 |
| | SL | 2011 | 9/20–12/31 | Low | 2613 | 2.59 | | 2.6 | 1.0 | 13.3 | 5.1 | 17.7 | 6.9 | 0.926 | 1.071 |
| | MM | 2013–15 | 1/1–12/31 | Mixed | 3424 | 3.12 | | NA | NA | NA | NA | NA | NA | NA | NA |
| 100HD | LL[3] | 2018–19 | 8/16–10/31 | Low | 2580 | 1.33 | NA | 7.2 | 5.4 | 14.9 | 11.3 | 22.5 | 17.0 | 0.89 | 0.980 |
| | LM[3] | 2018–19 | 1/1–10/31 | Mixed | 3310 | 1.78 | | 7.2 | 4.0 | 14.9 | 8.4 | 22.5 | 12.6 | NA | NA |

Observation stations are illustrated in Figure 1b, the first character in "Time Period" represents short-term (S), medium-term (M), and long-term (L), and the second character in "Time period" represents medium (M), high (H), low (L), or mixed (M) type flow scenarios. Superscripts 2 and 3 denote observation data used for comparison are from observation 2 and observation 3. $R^2$ and $\beta$ is a coefficient quantifying the degree of correlation between modeled and observed WSE and the slope of the linear regression of 1:1 plots. NA is used when observed data is not available.

in the upstream (E1 – E3) and the relatively straight downstream main channel (E9 – E10), but not good at the side channel with large curvature (E4 and E11). The agreement is reasonably good at main channels with big curvature (E5 – E8) and the outlet (E12). The corresponding correlation coefficients ($R^2$) between the CFD modeled and ADCP measured ones are 0.77 – 0.79, 0.75, 0.44 – 0.61, and 0.71 – 0.83, and 0.61 for E1 – E3, E9 – E10, E4/E11, E5 – E8, and E12, respectively. As $R^2$ of around 0.8 is usually recognized as an "acceptable" or "good" result in previous work (Nicholas and Sambrook Smith, 1999; Lane et al., 1999; Horritt, 2005; Lane et al., 2005), this means that the flow velocity predicted by the CFD model at most of the locations (9 out of 12) is reliable for practical applications. It is worth mentioning that the modeling accuracy for flow velocity may not be further improved by using more advanced CFD modeling or more refined mesh without improving the accuracy of ADCP and topography survey. For instance, Le et al. (2019) conducted a large-eddy-simulation for a 3.2 km long reach of the Mississippi River with a given discharge, the prediction accuracy of velocity was not improved when compared to ADCP measurements even though using 109 million grid and 38,400 CPU hours to reach a steady state. Furthermore, as the two dates chosen for velocity validation are randomly selected, it may be reasonable to expect that flow velocity modeling at other dates likely has similar accuracy, at least for short-term scenarios. This claim may be indirectly backed by the fact that WSE calibrated during 2011 still has similar accuracy as that in 2018 and 2019 (see Section 3.5).

### 3.4  Medium-term water stage validation

The short-term water stage validation shows the roughness calibrated using the WSE observed at a medium flow can well predict WSE at medium, high, and low flow scenarios. To further test if the calibrated roughness can be applied for medium-term (2 to 3 years after the calibration period) surface flow simulations, Figure 9 compares the modeled WSE with the observed WSE at 100B during 2013 – 2014. Figure 9a shows a comparison of the hourly recorded WSE from the model with those from two different observations. Such a comparison shows that the modeled WSE agrees well with the observations from January 1, 2013 to August 1, 2014. In addition, it shows that observed WSE has uncertainties. A further comparison between the two observations shows that WSE from observation 2 is about 3.2 cm higher than that from observation 1 and that a small shift in time results in a large error in standard deviation between the two observations (see uncertainty analyses in Appendix A2 and Figure A7). However, as observation 1 lacks the record during 2013 – 2014, observation 2 is used for validation during this time period.

As WSE observation is missing at some dates, three time periods with continual observations (see MH² and ML² in Table 2) were chosen to illustrate the modeling performance in predicting WSE as shown in Figure 9b,c,d. The comparison shows that the modeled WSE agrees very well with observations at the high flow scenarios during March–June 2013 (Figure 9b) and April–July 2014 (Figure 9d). The ME, MAE, and RMS during these periods are -10.1 cm – -9.2 cm, 10.5 cm – 11.9 cm, and 13.3 cm – 15.1 cm, respectively. The corresponding relative error to average water depth is -2.8% – -2.2%, 2.5% – 3.3%, 3.1% – 4.1%, respectively. At the low flow during September 2013-January 2014 (Figure 9c), the model shows a larger error especially when the WSE is low (close to 119 m). However, the relative errors to water depth, with values of -9.9%, 10.7%, and 12.6% for ME, MAE, and RMS (see ML² in Table 2), are still low. Figure 9e,f,g further shows a 1:1 comparison between

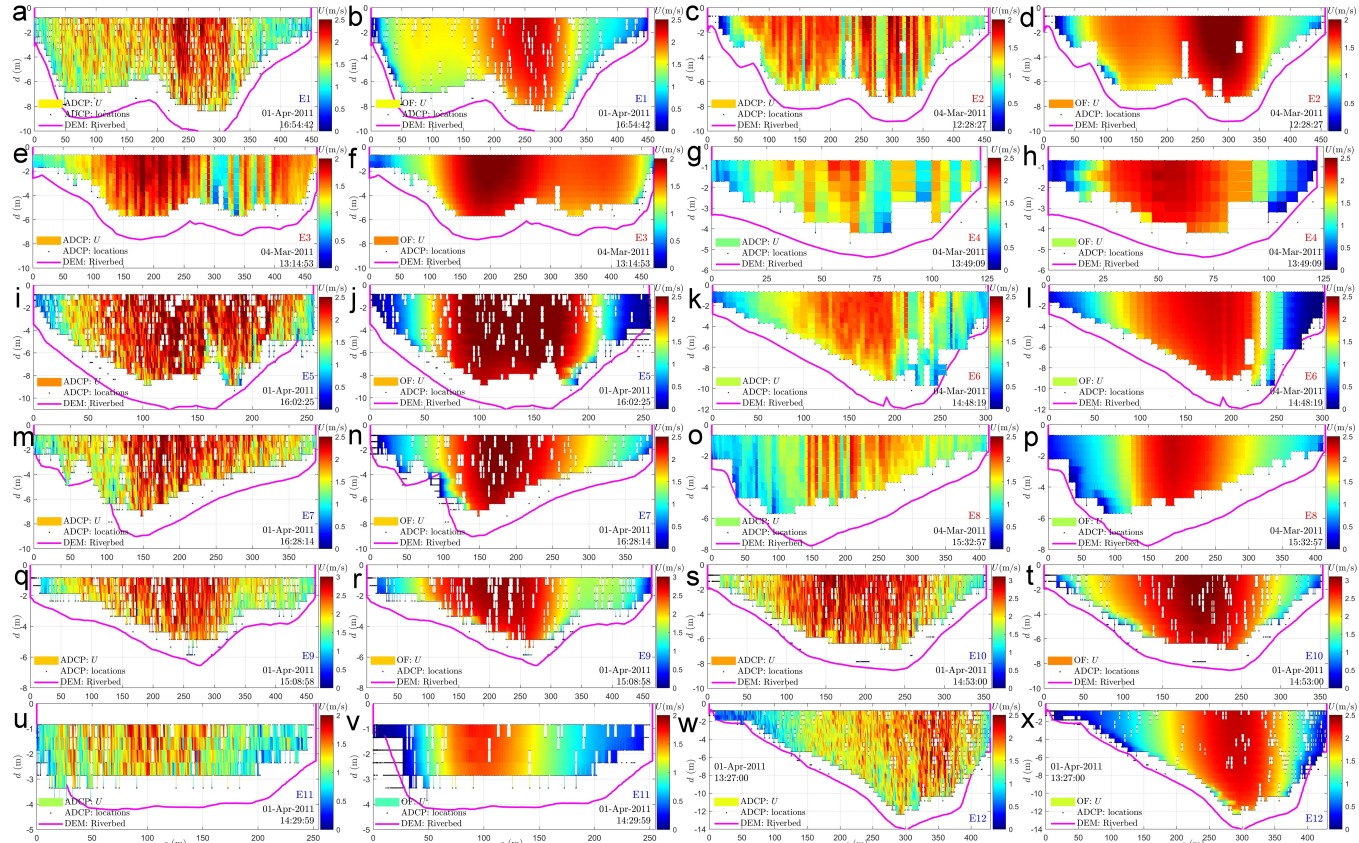

**Figure 7.** The velocity magnitude distributions on cross-section E1 – E12 from ADCP surveys (Columns 1 and 3) and CFD modeling (Columns 2 and 4). Cross-section names (E1 – E12) with red and blue color denote survey dates on March-4 and April-1, 2011, respectively. Symbols $d$ and $s$ denote depth away from the water surface and distance from the right bank (see Figure 1c), respectively.

modeled and observed WSE. The $R^2$ and the linear regression slope are 0.88 – 0.98 and 1.06 – 1.1, respectively. These results suggest the predicted WSE has no obvious bias and the prediction has good accuracy for a medium-term prediction.

### 3.5 Long-term water stage validation

The long-term (7 to 8 years after the calibration period) performance of WSE prediction is important for predicting river corridor function under a long-term climate change scenario. To test the modeling performance for long-term WSE prediction, Figure 10 compares the WSE from the model and the observation at one location (yellow dot in Figure 1b), different from the locations used for calibration. Figure 10a shows that the model well captures the trend of the fluctuation in WSE at 100HD during August 2018–November 2019. The ME and MAE are 7.2 cm and 14.9 cm, respectively. This is equivalent to 5.4% and 11.3% relative to the mean water depth. The RMS is 22.5 cm and about 17.0% relative to the average water depth at 100HD during August 2018–November 2019. Figure 10b further shows the 1:1 plot between the modeled and observed WSE

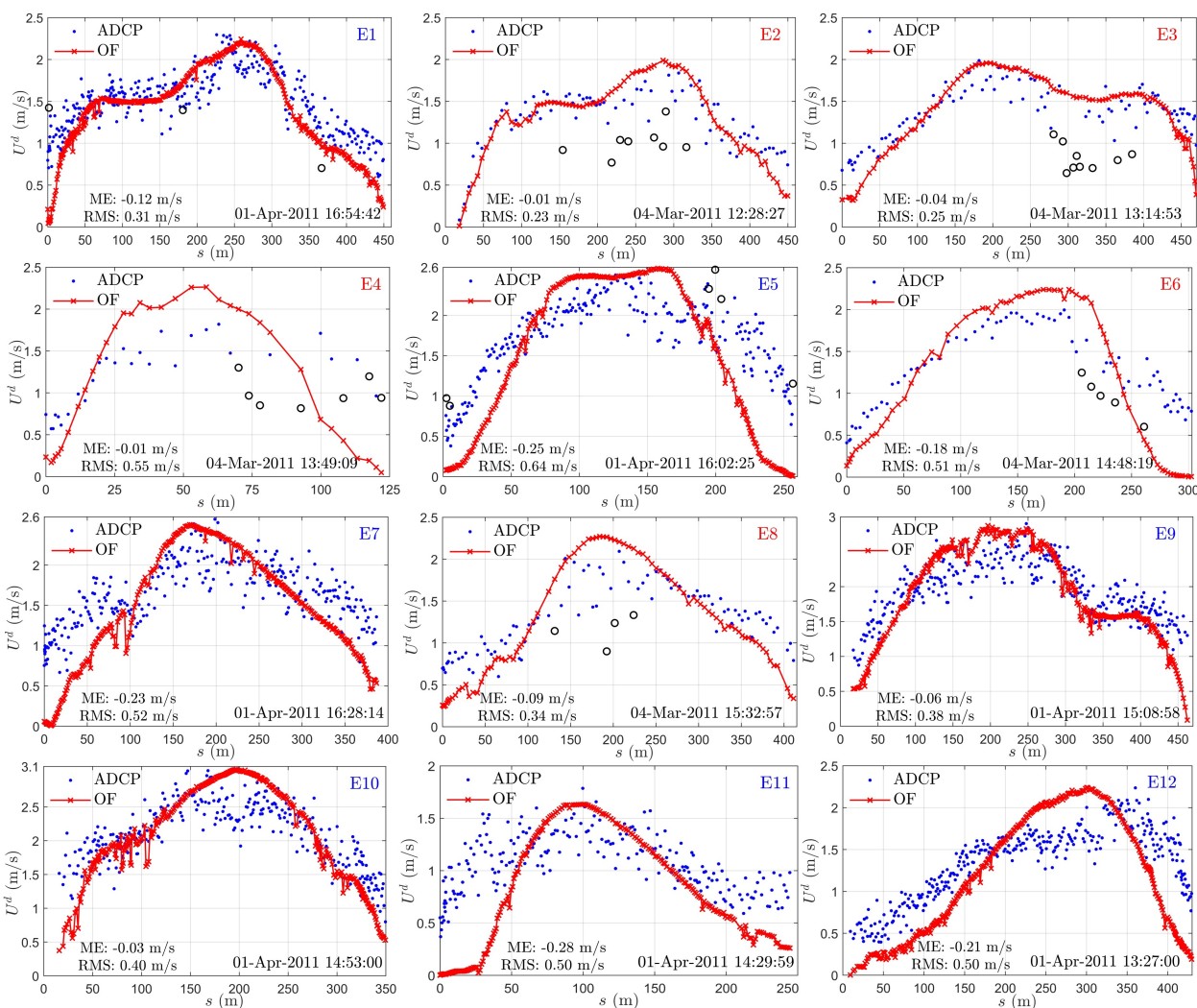

**Figure 8.** The comparison of depth-averaged velocity magnitude determined from ADCP surveys and CFD modeling. Black circles denote measured velocity outliers visually determined through velocity components (Figure A5 or A6).

at 100HD. The $R^2$ and linear regression slope are 0.89 and 0.980, respectively. These statistics show there is no obvious bias in our model as the slope is very close to 1. As the flow during August 2018–November 2019 is always low (2580 m$^3$/s), the $R^2$ during this time period is similar to those calculated at low flow scenario (see SL at 100B–100F in Table 2) in 2011–2015. Similarly, a lower $R^2$ is also related to a small time shift in the observation as shown in Figure A7. Considering that a small time shift in the observation results in a significant error in MAE and RMS, the ME is a more reliable index for evaluating the modeling accuracy. Therefore, it is reasonable to claim that our model can predict WSE in 2018 and 2019 with an accuracy of 5.4% relative to mean water depth using the roughness calibrated in 2011. This suggests that in the next 9 years the WSE may be reliably predicted using the calibrated roughness at the present time. As the water depth is another commonly used metric

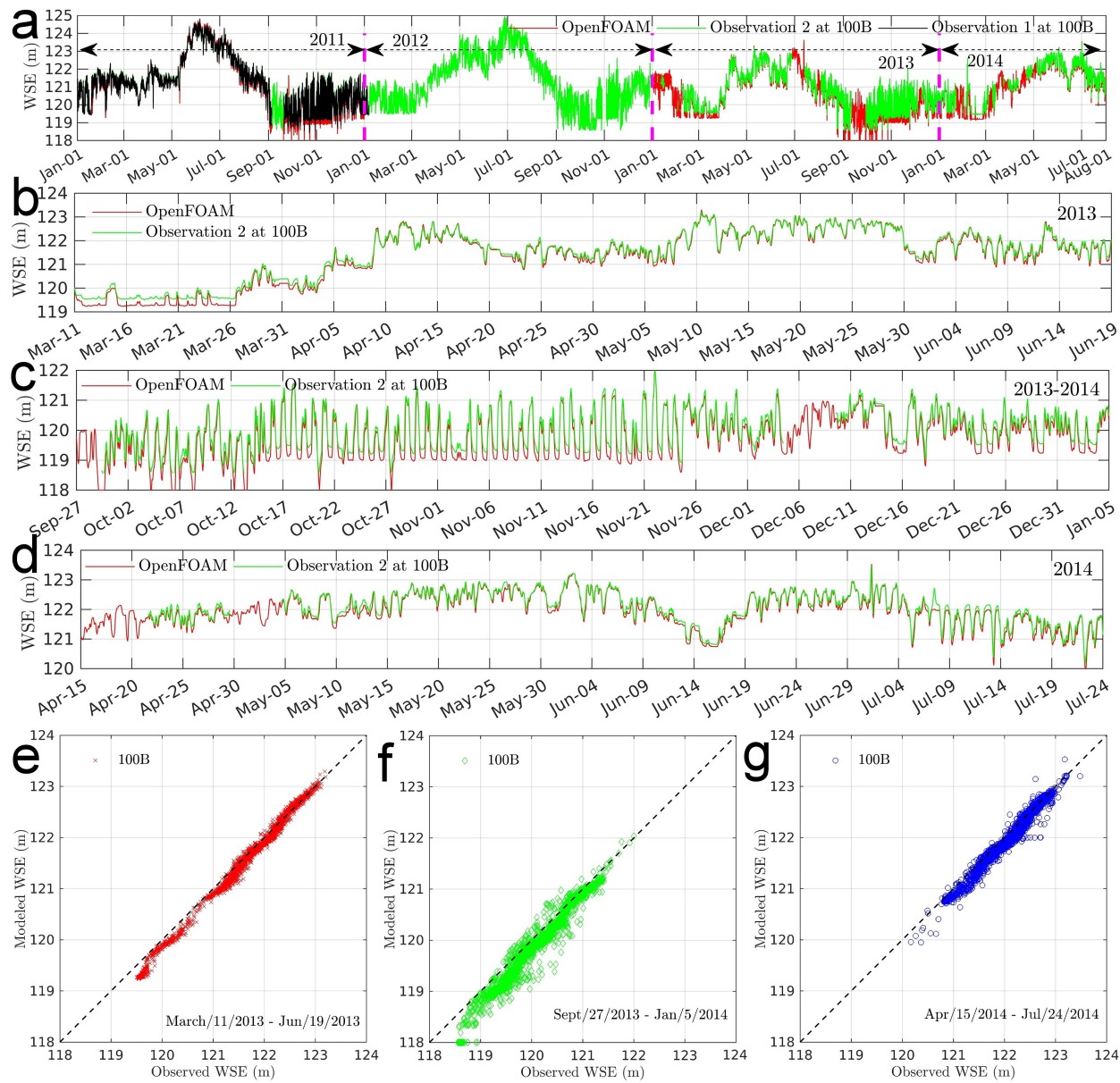

**Figure 9.** Medium-term model validation for water surface elevation. A comparison of hourly recorded WSE from model and observations during 2011 – 2014 (a), medium flow (b), low flow (c), and high flow (d). (e-g) denote the 1:1 plot during medium, low, and high flow scenarios.

for hydrodynamics model evaluation. Figures 10(c,d) show the comparisons of water depth from the model and observation at 100HD. It is observed that these results demonstrate identical visual and quantitative accuracy as those quantified by WSE.

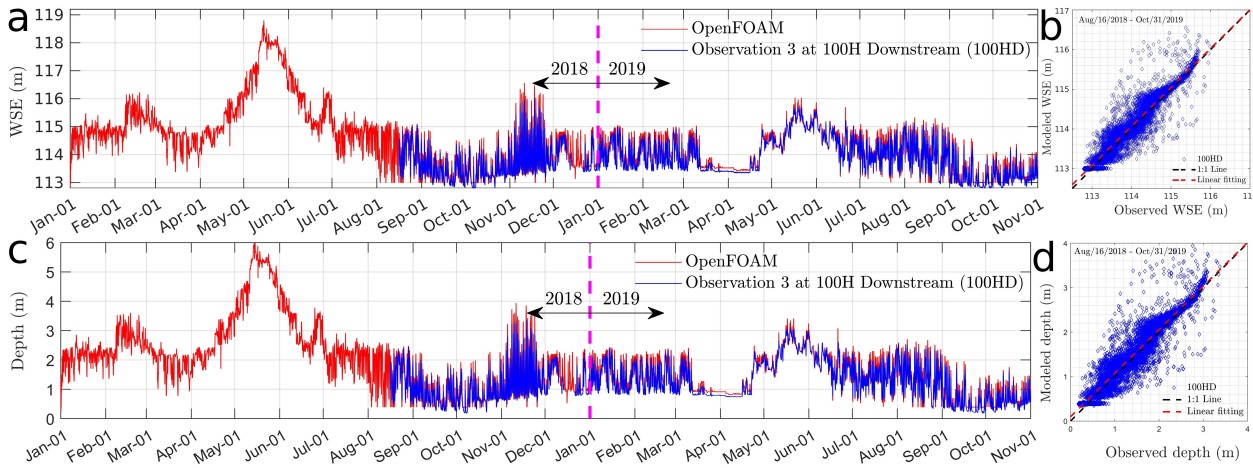

**Figure 10.** Long-term model validation for water surface elevation and depth. A comparison of the hourly recorded WSE and depth from model and observations at 100HD during 2018–2019 (a,c) and their 1:1 plots (b,d).

### 3.6   The ratio of dynamic pressure to static pressure

The dynamic pressure is important streamflow quantify, especially for environmental and ecological functions. However, modeling results of dynamic pressure in large-scale natural rivers are rarely reported and the relative importance of dynamic

pressure to hydrostatic pressure is also not clear. To quantitatively understand the relative importance of dynamic pressure to the hydrostatic pressure, we define their ratio as $r = p_d/[\rho g(\text{WSE} - z_b)]$ with WSE and $z_b$ denoting the water surface elevation and riverbed elevation. As such a ratio varies with location and discharge, we categorize the ratio into 5 ranges, including -0.4 – -0.3, -0.3 – -0.2, -0.2 – -0.1, -0.1 – 0, and 0 – 0.1. We then calculate the area ($A_r$) of which the pressure ratio falls in each range and its relative ratio to the total wetted area ($A_T$). Figure 11 shows the variations of the relative pressure ratio area

($A_r/A_T$) with time (a) and discharge (b), as well as the spatial distribution of each pressure ratio range at low (c), medium (d), and high (e) flow conditions. The results (Figure 11a) show that 60% – 80% of the riverbed is covered with dynamic pressure whose value is -10% to 0 of the hydrostatic pressure, while 10% – 30% of the total area is covered with dynamic pressure is of -20% to -10% of the hydrostatic pressure. The region with a dynamic pressure ratio higher than 0 or less than -20% is small. In addition, it was observed from Figure 11b that the relative pressure ratio area ($A_r/A_T$) behaves differently when the

discharge is less than 2000 m³/s (low flow), between 2000 m³/s and 4000 m³/s (medium flow), and large than 4000 m³/s (high flow), respectively. Specifically, the blue color is observed at both the dry-wet boundary and main channel at a low flow (Figure 11c), while it is mainly observed at the dry-wet boundary at a high flow (Figure 11e). At a medium flow, the blue area can be observed in both the main channel and the dry-wet boundary, though its area is obviously smaller than that observed in the low flow scenario. According to Figure 11a,b, the blue area could increase from around 10% at a high flow to around 30% at a

low flow. This means that dynamic pressure may be important at both the dry-wet boundary and the main channel at low flow conditions.

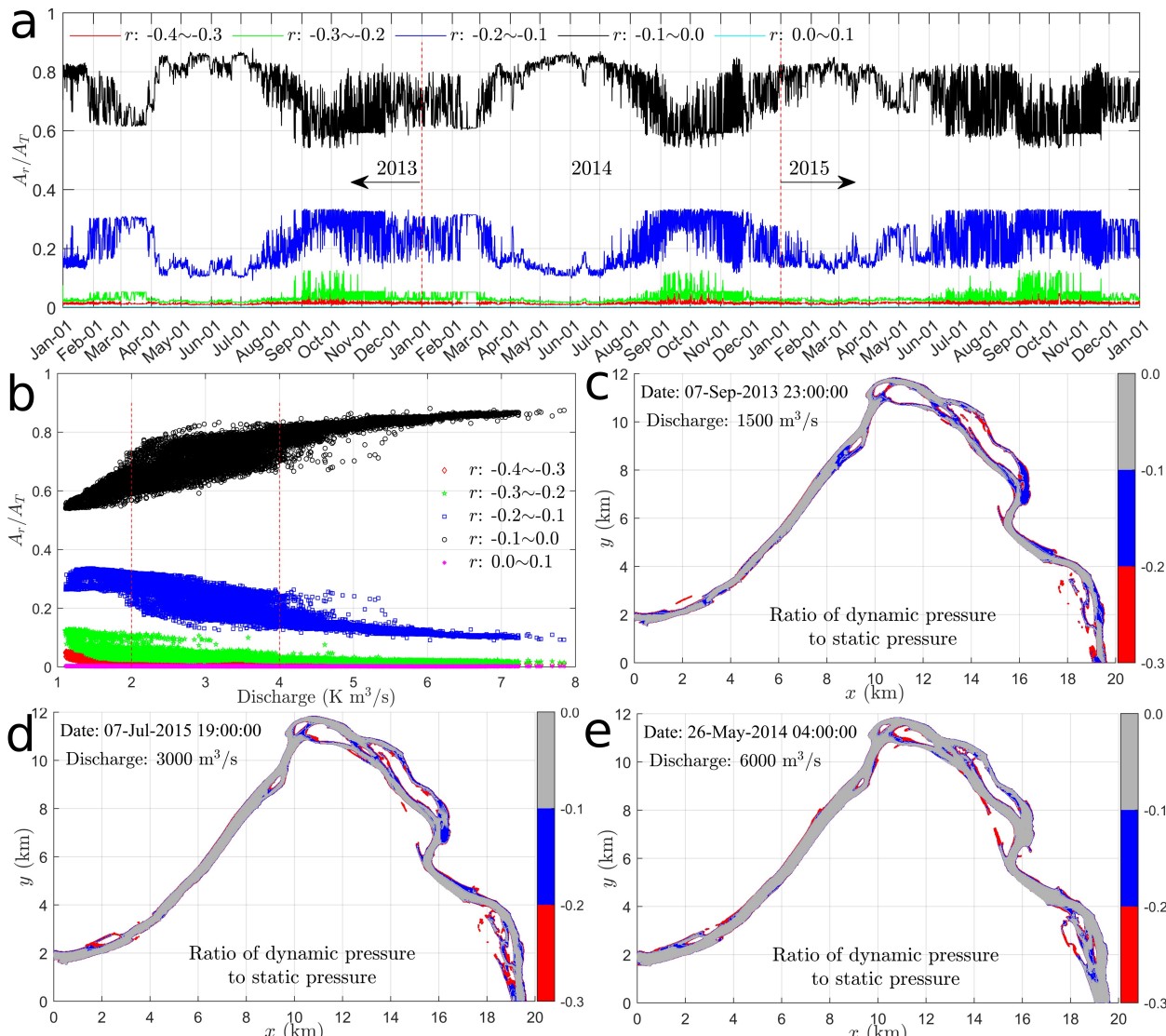

**Figure 11.** The variations of the relative pressure ratio area ($A_r/A_T$) with time (a) and discharge (b), as well as the spatial distribution of each pressure ratio range at low (c), medium (d), and high (e) flow conditions.

## 4 Discussion

### 4.1 Distributed hydraulic roughness estimation for large-scale rivers

Hydraulic roughness is a metric used to estimate the resistance applied to flow from complex sediment structures. Such a value controls the flow speed and water surface elevation, and has been long recognized as the primary control of the accuracy of numerical modeling of natural rivers (USACE, 1994). For small-scale rivers, assuming a uniformly distributed roughness is

usually acceptable. For large-scale rivers, however, it is necessary to use a distributed roughness height because the interactions between flow and local topographic features vary with locations. To guide roughness estimation in practical applications, we give an in-depth discussion on the roughness estimation approach used in the present work (Sections 4.1.1–4.1.2) and its
connections to other approaches such as Manning's coefficient (Section 4.1.4) and streambed microtopography (Section 4.1.5).

### 4.1.1  Calibration with observations: local optimal roughness height

Roughness calibration with observed water stage is an efficient approach for roughness estimation in 3D free-surface models. The physical basis of this approach is that the bulk flow velocity in streams is monotonically related to bed roughness and therefore an optimal roughness can be obtained by monotonically adjusting a roughness parameter to match modeled WSE
with observed ones. Usually, a very small roughness height, e.g., 0, results in an underestimation of WSE. While a high roughness height, e.g., the size of the biggest sediment, results in an overestimation of WSE. With this in mind, a series of numerical experiments can be designed by systemically adjusting the roughness parameter from 0 to the biggest value. And the relative error between modeled WSE and observed ones can be directly calculated as shown in Figure 3b. An optimal roughness parameter for each observation location can then be obtained, which is here referred to as a locally optimal roughness height.
Using such an approach, it is generally observed that the modeled WSE is very sensitive to the given roughness height when its value is much smaller than the optimal one (see Figure 3a,b, and Figures S2 and S3). For example, the ME increases by about 0.5 m – 0.7 m when the roughness height increases from 0 to 0.025 m (Figure 3b). However, further increasing the roughness height from 0.025 m to 0.05 m results in much smaller changes (0.1 m – 0.18 m) in WSE compared to that of changing from 0 to 0.025 m. These changes are even smaller when the roughness height approaches the optimal value. These
behaviors can be explained as follows.

Firstly, setting a zero roughness height is equivalent to using a smooth wall function (Versteeg and Malalasekera, 2007; CFDDirect, 2017). Such treatment is only valid when the shape, size, and distribution of individual sediments on the riverbed are explicitly represented by the riverbed topography. For almost all CFD modeling of natural rivers, however, the details of individual sediments cannot be measured as the commonly used survey technology, i.e., LiDAR, cannot capture geometric
details smaller than a half meter (Podhorányi et al., 2013; Tonina et al., 2019). Therefore, setting a zero roughness height on top of a LiDAR-measured topography results in large errors in predicting WSE when compared to observed ones. By contrast, using a non-zero roughness value considers the effects of the missing geometric details on flow, which makes the model more approaching to the real situation. This can be demonstrated by similar values of the optimal roughness heights, i.e., 2.83 cm – 25.56 cm (Table 1 Case OF0), to typical sizes of gravel and cobbles (2 mm – 0.256 m) (Berenbrock and Tranmer, 2008).
Hence, it can be concluded that the roughness wall model and non-zero roughness heights reduce the sensitivity of WSE to roughness height; and they provide a reliable mechanism for roughness calibration. It is worth mentioning that the sensitivity of WSE to roughness height can be further reduced if details (mm-scale) of individual sediments on riverbed can be measured and explicitly represented by sufficiently small (mm-scale) mesh in the CFD model (Lane et al., 2004; Hardy et al., 2005). However, measuring a river topography and generating a mesh with mm resolution is currently impractical for large-scale
natural streams. Our approach discussed here, therefore, is still of great practical importance.

### 4.1.2 Calibration with observations: local roughness adjustment

As the roughness parameter calibrated in Section 4.1.1 usually works well for a single location, this means that applying such a parameter to other locations cannot guarantee overall modeling accuracy for all locations. Different strategies can be applied to solve this problem. The simplest strategy is to choose one roughness parameter and apply it uniformly to the whole domain. Such a parameter can be directly identified from error diagrams (Figure 3b or Figure A3), which has a value of $k_s^1$ = 12.2 cm. Using this strategy, the overall modeling accuracy is about -30 cm – 30 cm and 7.5 cm – 30 cm in terms of ME and MAE (see OFK1 in Table 1). The second strategy is to decompose the riverbed into two regions with different roughness parameters assigned to each region. This strategy is based on the fact that the error diagrams (Figure 3b or Figure A3) show two different behaviors at the region 100B and the other five locations. Following this concept, $k_s^{2b}$ = 25.56 cm is assigned for the region at 100B and $k_s^{2a}$ = 6.25 cm is assigned for all other regions. The overall modeling accuracy for WSE using such a strategy is around -17 cm – 15 cm and 9 cm – 15 cm in terms of ME and MAE (see OFK2 in Table 1). Overall, we see that the modeling accuracy of using one or two roughness values is $\pm$ 0.3 m and $\pm$ 0.15 m in terms of ME, and 0.3 m and 0.15 m in terms of MAE. It is important to mention that such a modeling accuracy can be roughly predicted using error diagrams without running actual simulations (cases OFK1 and OFK2 in Table 1). This means that the error diagram is a good tool for designing a calibration strategy. We also tested the strategy of interpolating the locally optimal roughness height to 50 uniformly distributed regions (see $k_s$ and regions in Figure A8). The overall accuracy for WSE is -19.4 cm – 8.5 cm and 9.3 cm – 19.4 cm in terms of ME and MAE (case OFK50 in Table 1), respectively. This result suggests that interpolating the locally optimal roughness height to more regions does not improve modeling accuracy because roughness interpolating itself may introduce extra uncertainty to the roughness field. From the above discussion, we found that the best strategy is to decompose the riverbed into N regions with N equal to the number of survey locations. Without further adjustment of the local optimal roughness parameters, such a strategy gives an overall modeling accuracy of WSE as -16.5 cm – 6.4 cm and 7.6 cm – 19.6 cm in terms ME and MAE, respectively.

To further improve the modeling accuracy, local adjustment of the local optimal roughness parameters is necessary. This is because the locally optimal roughness parameters neglect the flow interactions due to locally variable flow resistance, backwater effects from downstream to upstream, and the effects of sinuosity. The local adjustment is used to incorporate these effects into the calibration and achieve a globally optimal roughness calibration. As higher uncertainty (case OF0 in Table 1) occurs at the upstream locations (100B, 100N, and 100D) using the locally optimal roughness height, we systematically adjust the roughness parameters at these locations. The final modeling accuracy for WSE is -7.5 cm – 6.4 cm and 7.5 cm – 12.7 cm in terms of ME and MAE, respectively. Further improvement of the accuracy is possible but not necessary as the relative errors to water depth have been reduced to -2.7% – 2.1% and 2.1% – 5.3% in terms of ME and MAE.

Nevertheless, it is worth summarizing how local adjustment improves modeling accuracy. Firstly, increasing roughness height at the most upstream location (100B) improves the accuracy of WSE only at that location (see OF0, OF1, and OF2 in Table 1). Secondly, changing roughness height at 100N has little effect on WSE at 100N and neighboring upstream locations (see OF2 and OF3 in Table 1). And thirdly, increasing roughness height at 100D significantly affects WSE at all upstream

locations and has a larger influence on the locations closer to that location. These results suggest that roughness heights at some critical locations (most upstream and close to pool) have a larger impact on the overall modeling accuracy. It is also worth noting that the calibration strategy discussed in Sections 4.1.1–4.1.2 is designed based on water stage measurement data availability. Such a strategy can provide unique roughness values in the local optimal roughness calibration step, however, may not provide unique values in the local roughness adjustment step. Considering that the overall model performance is controlled

by both steps, future applications are recommended to pay more attention to the first step due to its uniqueness. Specific actions include deploying stage survey devices at critical locations (upstream, pools, bends, islands, etc.) and in characteristic sediment environments, e.g., gravel, sand, mixed gravel and sand, vegetation, etc. With a better distribution of the stage survey locations, the overall model accuracy solely based on the first step is likely significantly improved, consequently the local roughness adjustment step becomes less important. If the adjustment step is a must, integrating the present CFD framework with machine

learning approaches, e.g., parameter learning (Tsai et al., 2021) and reinforcement learning, can potentially address the non-uniqueness issue.

### 4.1.3 Sensitivities of water surface elevation and velocity magnitude variations to roughness heights

Though the above roughness estimation strategy only targets minimizing the water stage differences between the model and observations at six locations, it is worth noting that the roughness values also affect the spatial variations of water stage and

velocity. By selecting the Case OF (see Table 1) as a reference case, Figure 12 compares the distributions of water stage and free surface velocity magnitude from the reference case with those cases using different uniform roughness heights at two transects L1 and L2 (see Figure 1c). In particular, Figure 12a shows that the average WSE at transect L1 increases from 118.4 m to 120 m with the roughness height increasing from 0 m to 0.5 m. This contributes to only an 18% increase in the water depth at the channel valley (elevation 109.4 m), however, it significantly affects the model accuracy in predicting the water

depth near the river banks where the submerged bed elevation from the reference case is around 119 m. At transect L2 (Figure 12d), increasing the roughness height from 0 to 0.5 m causes an increase in WSE from 113.9 m to 115.7 m if selecting the black circles as the average WSEs. This means the water depth at the channel valley increases 28% (bed elevation 107.55 m) and also causes significant changes in the water depth near the banks whose elevation is around 115 m. Figure 12d also shows that the spatial variation of WSE is affected by the selection of roughness parameters, especially at a lower value. In summary, the

water depth near the river banks is highly sensitive to the roughness values while the sensitivity gets reduced when approaching the river valley. Due to such a reason, the model calibration should pay more attention to the comparison of water stage from model and observations at river bank regions.

Because velocity at the water surface is a good indicator of the velocity distribution below the surface, Figure 12b compares the distribution of surface velocity magnitude at transect L1 under different roughness heights. The result shows that the

velocity distribution from the zero roughness case differs significantly from the other cases with non-zero roughness heights. For the cases with non-zero roughness heights, increasing the roughness value (from 0.025 m to 0.5 m) does not significantly affect the spatial distribution of velocity, however, reduces its maximum value from 2.46 m/s to 1.92 m/s (22% decreasing). The relative error ($r_U$) of the maximum velocity between these cases and the reference case for L1 is shown in Figure 12c from

which the relative error is observed to decrease from 10% to around -15% with increasing roughness heights. Similar behaviors can be observed at L2 as shown in Figure 12f. With the roughness height increasing from 0.025 m to 0.5 m, the maximum velocity is reduced from 2.68 m/s to 1.88 m/s while the relative error $r_U$ decreases from around 10% to -20%. From these results, we conclude that the distribution of surface velocity is not sensitive to the non-zero roughness heights but its maximum value could vary 25% – 30% for the roughness range considered here.

Interestingly, it is observed that the velocity distribution of the reference case (Figure 12b,e red line) falls in between the velocity distributions of cases with heights 0.05 m (Figure 12b,e dashed blue line) and 0.1 m (Figure 12b,e dashed cyan line) while the water stage of the reference case (Figure 12a,d red line) is falling between the ranges of the same roughness cases (Figure 12a,d dashed blue and cyan lines). This indicates that the roughness height calibration with water stage is equivalent to calibrating it against water surface velocity. Noting that the water stage is highly sensitive to roughness at river banks while the water surface velocity is more sensitive to roughness at the river valley, future model development may consider calibrating distinct roughness values for river banks and valleys using a combined water stage and surface velocity calibration.

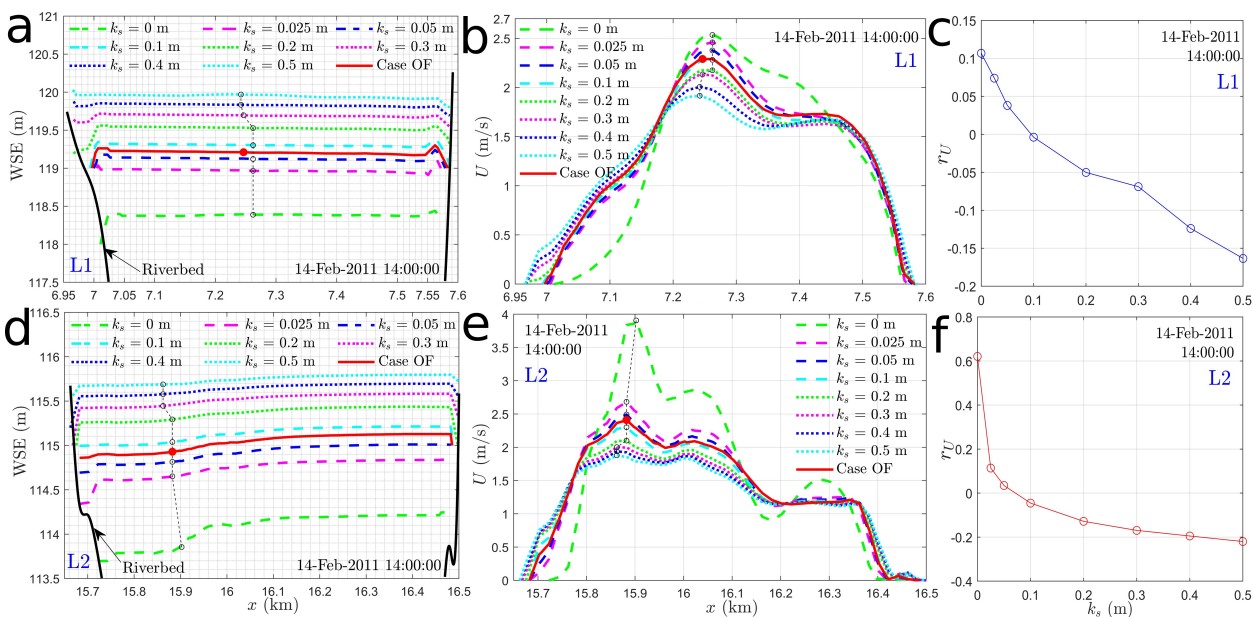

**Figure 12.** The spatial variations of water surface elevation (a,d) and free surface velocity magnitude (b,e) under different roughness heights at transects L1 and L2 (see Figure 1c) as well as the relative error (c,f) of the maximum free surface velocity at each roughness height. Dashed circle lines and red dots in a–b and d–e denote the locations where the maximum velocity magnitude is reached. The $r_U$ in (c,f) is defined as the relative velocity difference between the black circles and red dots in (b,e).

### 4.1.4 Converted from Manning's coefficient

Though the above roughness calibration approach can be applied for any rivers where WSE observation is available, such a process is usually time-consuming. 1D and 2D models have been widely used to predict WSE and Manning's coefficients have been available in these models. For example, for the river section studied in this work, the calibrated Manning's coefficients from a 2D CFD model, MASS2, are 0.038, 0.035, 0.034, 0.027, 0.027, and 0.03 (with unit s/m$^{1/3}$) at 100B, 100N, 100D, LI, 100H, and 100F (Niehus et al., 2014). In these situations, the roughness parameter required in 3D CFD models can be directly converted from the well-calibrated Manning's coefficients based on a force balance at the riverbed. Specifically, the force balance can be described as $\tau_b = \rho g S R = 1/8 f \rho U^2$ with $\tau_b$, $S$, $R$, $f$, and $U$ denoting average bed shear stress, channel slope, hydraulic radius, Darcy-Weisbach friction factor, and average streamwise velocity. For gravel-bed rivers, it was shown that $\sqrt{\frac{8}{f}} = a(\frac{R}{k_s})^b$ with $b = 1/6$ and $a$ has a value of 6.7, 7.3, 8.2, 8.4, 9.39, etc. when $R/k_s > 10$ (Chaudhry, 2008; Rickenmann and Recking, 2011; Ferguson, 2019). Meanwhile, the Manning's equation shows $U = \frac{1}{n} R^{2/3} S^{1/2}$ with $n$ denoting the Manning's coefficient. Using these formulas, the relationship between $n$ and $k_s$ can be quantified as $n = \frac{1}{a\sqrt{g}} k_s^{1/6}$ if $k_s$ has a unit of foot or $n = \frac{1.219}{a\sqrt{g}} k_s^{1/6}$ if $k_s$ is in SI unit. The coefficient $a$ characterizes the type of sediment that requires further calibration, however could use an average value of 8.0 for a rough estimation of $k_s$. In this work, as the locally optimal roughness height can be deterministically calculated and the modeled WSE at 100F gives a very good accuracy (see 100F in OF0 Table 1), we back-calculated the value of $a = 8.4$ using $k_s$ = 7.42 cm = 0.2434 ft and $n = 0.03$ s/m$^{1/3}$. With the calibrated value for $a$, hydraulic roughness $k_s$ can be converted as shown in case MS in Table 1. The modeling accuracy of WSE using these roughness parameters is -4.7 cm – 7.7 cm and 6.4 cm – 13.9 cm in terms of ME and MAE, respectively. This result suggests that the roughness height converted from the well-calibrated Manning's coefficients of 2D models can give similar modeling accuracy compared to using the globally optimal roughness height. Further local adjustment of these roughness parameters does not significantly improve modeling accuracy (see MS2 and MS3 in Table 1).

### 4.1.5 Estimated from microtopography

Both roughness calibration and conversion from the Manning's coefficients require observation of the water stage and these calibrations may not guarantee the accuracy of other flow quantities such as bed shear stress and velocity. A more accurate and physics-based method for evaluating the effects of bed roughness is to directly resolve the influence of microtopography on flow dynamics. However, the success of such a method depends on high-resolution measurements of riverbed microtopography, computational techniques capable of resolving complex geometry in CFD codes, and available high-performance computing resources. Owing to the rapid development of structure-from-motion (SfM) photogrammetry and unnamed aerial vehicles, remote sensing of riverbed sediment structure with 1 cm – 5 cm resolution over a 40-kilometer river reach has been possible for shallow streams (Carr et al., 2019). SfM survey of a patch-scale (5 m$^2$) natural streambed 0.5 m beneath water surface has also been recently realized with a 1 mm resolution (Danhoff and Huckins, 2020). These data can be used either for quantifying locally distributed grain size distribution or used as a geometric boundary for 3D CFD models where the effects of sediment structure on flow dynamics can be directly resolved. At the patch scale (a few meters to tens meters), SfM photogrammetry-

scanned high-resolution (mm – cm scale) natural riverbeds have been used to directly resolve the effects of sediment structure

on the flow resistance (Chen et al., 2019). A quantitative relationship has been identified between hydraulic roughness height, turbulence vortex structure, and characteristic sediment size. Therefore, with available high-resolution riverbed structures from SfM and existing theories on hydraulic roughness, the distributed hydraulic roughness height in large rivers can be directly estimated and integrated with the CFD code.

## 4.2 OpenFOAM medium and long-term water stage prediction performance compared to 1/2D models

Though Section 4.1 discusses the roughness estimation procedure for a short time, we want to emphasize that the procedure and the usage of the roughness wall model are key to maintaining the model's accuracy over long time period and large spatial extent. Their importance can be illustrated by comparing the WSE from MASS1 (Richmond and Perkins, 2009), MASS2 (Niehus et al., 2014), OpenFOAM, and observations as shown in Figure 13 and Table A3. Here, the three models are calibrated with WSE during similar time periods (October 2010 – March 2011) using the same river topography, discharge, and stage

data. The calibration accuracy of these three models are -0.2 cm – 0.2 cm, -3.8 cm – -0.8 cm (Table A3), and -7.5 cm – 6.4 cm (Table 1 Case OF) in terms of ME; and 4.8 cm – 17.6 cm, 3.9 cm – 12.8 cm, and 7.5 cm – 12.7 cm in terms of MAE. These data demonstrate that the 1D (MASS1) and 2D (MASS2) models were calibrated to better accuracy than the 3D model during the calibration period. Using this calibrated roughness, Figure 13a compares the WSE from the three models and the observation at location 100B during April to June in 2013 (in medium-term). The result suggests that the 1D and 2D models

overestimate the WSE of about 0.4 m, while the 3D is still very accurate at most dates, even though the 1D/2D models have a better calibration accuracy. Further, to examine these models' long-term predictive capability at locations outside calibration locations, Figure 13b compares the WSE from these models and another observation at location 100HD during 2018 and 2019 (long-term). The result shows that the WSE predicted by the 1D model deviates from that from the 2D/3D models and observations. Such a deviation can be more clearly observed through the 1:1 plot between modeled and observed WSE (Figure

13c). Figure 13c also shows that the WSE from the MASS2 and OpenFOAM has no obvious bias relative to the observation data. Figure 13d further shows the 1:1 plot between modeled WSE from MASS1/MASS2 and OpenFOAM, which clearly suggests that WSE from MASS2 has similar accuracy as OpenFOAM, but MASS1 deviates from it, especially at the lower WSE (low flow conditions). From these results, it is reasonable to conclude that the 3D CFD model framework proposed in this work can reliably predict WSE over short-, medium-, and long-term periods at both calibration and non-calibration locations.

The 1D and 2D models, though with accurate calibration, may not maintain its predictive capability for medium and long-term streamflow at some locations. The lower accuracy of 1D/2D models may be attributed to their intrinsic physical simplifications, e.g., cross-sectional or depth average and resulting nonphysical meaning of roughness parameter (Lane et al., 2005; Lane and Ferguson, 2005), which necessitate re-calibrating bed roughness to account for the dynamic changes in discharge.

## 4.3 Effects of discharge variations and topography heterogeneity on riverbed dynamic pressure

Riverbed dynamic pressure gradient affects the water exchange between the stream water and groundwater. Depending on the treatment of dynamic pressure, existing surface-subsurface interaction models can be categorized into three types: no dynamic

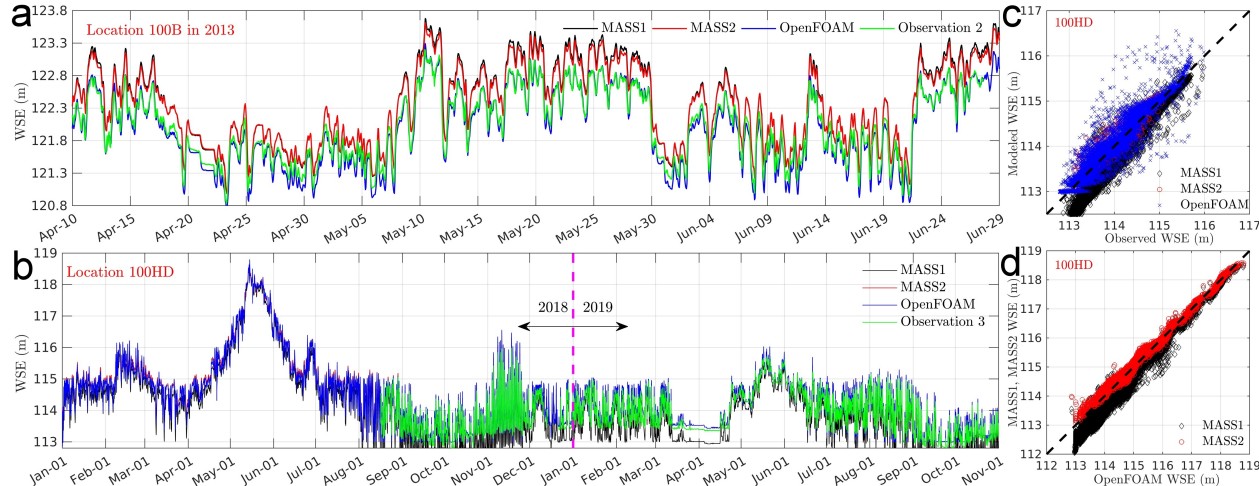

**Figure 13.** A comparison of water surface elevation (WSE) from MASS1, MASS2, OpenFOAM, and observations at 100B during 2013 (a) and at 100HD during 2018 - 2019 (b). (c) denotes the 1:1 plot of WSE between models and observation for 100HD; and (d) denotes that between OpenFOAM and MASS1/2. Details of the WSE from MASS1 and MASS2 can be found in (Niehus et al., 2014).

pressure models, one-way coupled dynamic pressure models, and two-way coupled dynamic pressure models. The models that solve the surface water using 1D/2D Saint-Venant equations and 3D hydrostatic Navier-Stokes equations belong to the first type because dynamic pressure is ignored (Maxwell et al., 2014). The second type solves the surface water using a 3D hydrodynamic model and then uses the computed riverbed total pressure as a boundary condition for the subsurface flow without feeding the fluxes from the subsurface back to the surface water (Cardenas and Wilson, 2007; Bao et al., 2018; Zhou et al., 2018). The third type is similar to the second type but allows the fluxes from the subsurface back to surface water (Broecker et al., 2019; Li et al., 2020). Though the third-type of models are desired to study the impact of dynamic pressure gradient on surface-subsurface interactions, they are currently limited to laboratory scale and idealized flow conditions due to high computational costs and model instability. For the spatiotemporal scales studied in this work, the one-way couple approach is the only solution. Despite the importance of the dynamic pressure, existing one-way coupled models usually neglect the effects of dynamic pressure based on an assumption that the dynamic pressure is negligible compared to the hydrostatic pressure. With the CFD modeling results reported in Section 3.6, it is found that the relative importance of dynamic pressure to hydrostatic pressure varies with discharge and riverbed topography. In general, the dynamic pressure is less than 10% of the hydrostatic pressure in 60% to 80% of the total wetted area and is between 10% and 20% of the hydrostatic pressure in 10% to 30% of the region. With variations in discharge, 20% more area could be covered by higher dynamic pressure (10% and 20% of the hydrostatic pressure) at low flow (< 2000 m$^3$/s) compared to that at a high flow (> 4000 m$^3$/s). Spatially, both the main channel and dry-wet boundaries (shorelines and island boundaries) are likely covered with the above higher dynamic pressure at a low flow. While only the dry-wet boundaries are covered with the higher dynamic pressure at a high flow. As 30% of the wetted area could be covered with dynamic pressure whose value is 10% to 20% of the hydrostatic pressure, whether it is acceptable to neglect the dynamic

pressure in existing surface-subsurface models is questionable. In addition, the frequent discharge fluctuations cause variations in the magnitude and coverage area of the dynamic pressure. These dynamic variations likely further affect the water exchange rate between stream and groundwater. The specific impacts of riverbed dynamic pressure on the surface-subsurface exchange have been reported in our another work (Bao et al., 2022).

580 ## 4.4   Computational efficiency

Despite the rapid growth in computational capacity in the past three decades, it is still a bottleneck for CFD modeling of natural rivers with tens of kilometers scale over multiple years. However, we show that such a limitation may be relieved using highly efficient CFD code, spatiotemporal decomposition approach, and a few hundred CPUs commonly available in university-scale or national-scale cyberinfrastructure. The discussion here is based on modeling results during 2011 (1 month), 2013-2015 585 (36 months), and 2018–2019 (22 months) by using Cascade, a high-performance computer managed by the Environmental Molecular Sciences Laboratory (EMSL) at PNNL (www.emsl.pnnl.gov). For convenience, we define wall-clock time, CPU time, and solution time as the real-world time experienced by human, the time consumed by the computer, and the time of water flow in the CFD model, respectively. With these definitions, the computational efficiency can be quantified by the ratio of solution time to wall-clock time.

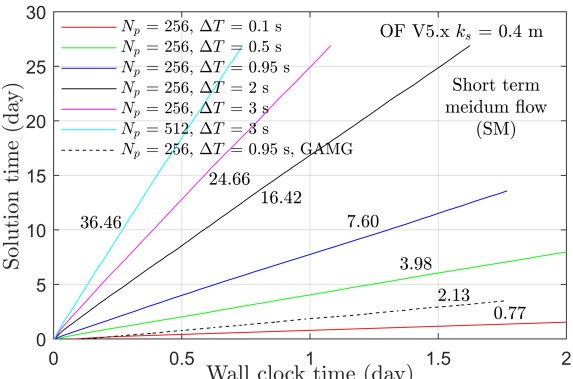

**Figure 14.** The advancement of solution time with respect to wall-clock time. $N_p$ and $\Delta T$ denote the number of processors and time step. The linear solvers represented by solid and dashed lines are DIC-PCG and GAMG, respectively. The line slope or the computational efficiency is denoted by the values adjacent to each line.

590     Figure 14 shows the advancement of solution time with respect to wall-clock time for the short-term medium flow case. It is observed that the computational efficiency, i.e., the slope of each line, increases linearly with increasing time step $\Delta T$ (solid lines with processor number $N_p = 256$). In addition, increasing the number of processors from 256 to 512 only increases the computational efficiency by 1.5 times (see magenta and cyan lines). Further increasing the number of processors decreases the computational efficiency, which means that an optimal number of processors, i.e., $N_p = 512$, exists for our model. The 595 computational efficiency is also affected by the selection of the linear solver. In our case, the PCG solver with DIC conditioner

increases the computational efficiency by 3.6 times compared to using a generalized geometric-algebraic multigrid (GAMG) solver (see blue and dashed black lines). Despite the changes in time step and number of processors, the modeled WSE does not change (see Figure A4). Following these analyses, we show that the computational efficiency is around 36 by using 512 processors, 3 s as the time step, and DIC-PCG as the linear solver. This means we can simulate 1 month solution time in less than one day of wall-clock time or one day solution time in 40 minutes (1/36 days) of wall-clock time. With the same parallel computation setups, we divide simulations during medium-term and long-term into 36 and 22 cases and run all cases simultaneously. This approach does not reduce the total CPU time, but significantly reduces the maximum wall-clock time required to complete all simulations. The OpenFOAM log files (see Data sets) show that all simulations were completed in less than 6 days of wall-clock time. Considering the number of processors, the total CPU hours spent is about 1.1 million, which is equivalent to 19,000 CPU hours for each month. Note that the time considered here does not include the computational time used for calibration which is around 28% of the total CPU hours. However, our work shows that calibration is only required once. Therefore, for rapid predictions of the streamflow with well-calibrated roughness parameters, the computational efficiency is likely feasible in terms of how much time and how many CPU hours are required.

## 5   Conclusions

This work proposed a semi-automated workflow that combines topographic and water stage surveys, 3D computational fluid dynamics modeling, a distributed rough wall resistance model, and spatiotemporal decomposition to simulate the streamflow in a 30-kilometer-long river reach in the Columbia River spanning 5 years. Specifically, a LiDAR measured river topography is represented by a zig-zag grid in the 3D model. The effect of geometric differences between an actual riverbed and the computational mesh on streamflow is modeled with a distributed rough wall resistance model with the roughness parameters calibrated with measured WSE at six locations during 2011. The time decomposition approach enables decomposing the simulation period 2013-2015 into 36 months and 2018-2019 into 22 months with each month simulated simultaneously using parallel computation. Further computational efficiency analyses show that the time step, number of processors, and selection of linear solver affect the final computational efficiency. Using the spatiotemporal decomposition approach, the 3D CFD modeling of the streamflow in 58 months can be achieved in less than six days with a cost of 1.1 million CPU hours.

Systematical roughness calibration shows that the distributed roughness field enables an average WSE difference between modeled and observed ones as -7.5 cm – 6.4 cm, which is equivalent to -2.7% – 2.1% relative to average water depth. With this calibrated roughness field, the modeling accuracy for WSE is reported as -15.6 cm – 9.1 cm, -14.4 cm, and 7.2 cm for short-term, medium-term, and long-term predictions, which is equivalent to -7.1% – 6.6%, -4.6%, and 5.4% relative to the average water depth. The model also demonstrates its predictive capability in reproducing the flow distribution and depth-averaged flow velocity at 9 out of 12 cross-sections with correlation coefficients 0.71 – 0.83. Using the validated modeling results, the relative importance of dynamic pressure to hydrostatic pressure and its dependencies on discharge variations and topography heterogeneity are further studied. It is found that the dynamic pressure is less than 10% of the hydrostatic pressure in 60%

to 80% of the total wetted area while it is 10% to 20% of the hydrostatic pressure in 10% to 30% of the wetted region. The relative importance and the coverage area are found to change with discharge and locations.

Given the high modeling accuracy and computational efficiency of our model, this work provides a generic framework to evaluate and predict the impact of climate- and human-induced discharge variations on river flow velocity, stage, and dynamic pressure at decade temporal scales and tens kilometer spatial scales that are relevant to the hyper-resolution (0.1 – 1 km) global- and continental-scale land surface (Wood et al., 2011; Bierkens et al., 2015) and groundwater modeling (Condon et al., 2021). With the discharge from global hydrological models at relevant scales, e.g., 5 to 10 km in space and hourly to daily in time

(Lin et al., 2019; Alfieri et al., 2020; Harrigan et al., 2020; Yang et al., 2021), the streamflow model can be better constrained by climate- and human-induced discharge disturbances and can also serve as a testbed for the characterization of the processes at the scales (less than 0.1 km) not represented in global hydrologic models.

## Appendix A: Uncertainty analyses

### A1    Mesh resolution and time step uncertainty

The mesh resolution and time step are common sources of uncertainty of CFD models. As one goal of this paper is to predict the total pressure at the streambed, a summation of the hydrostatic pressure and the dynamic pressure, Figure A1 shows the difference and the 1:1 plot of the total pressure head between a fine mesh (20 m $\times$ 20 m $\times$ 0.5 m) and a coarse mesh (20 m $\times$ 20 m $\times$ 1 m) at the time 16PM Jan-16-2013. The result shows that the difference is in the range -0.1 m – 0.1 m at most of the locations and the spatial average difference is -0.03 m (Figure A1a). The 1:1 plot also shows that the total pressure head from

the two meshes almost overlaps with a mean difference, a root mean square, and a $R^2$ value as -0.03 m, 0.1 m, and 0.9987, respectively (Figure A1b). Recalling that the WSE (related to the hydrostatic pressure head) observation itself could have an uncertainty of 0.032 m (see Appendix A2), the uncertainty attributed to mesh resolution is of the similar order of uncertainty in water stage observation. This suggests that the mesh resolution does not contribute significant error to the total pressure head. To further evaluate the effect of time step, Figure A4 shows a comparison of the modeled WSE using five different time steps at

the six observation locations. The results reveal that the time step tested here does not affect the accuracy of WSE. Therefore, we choose the time step 3 s as the final time step in order to reduce computational costs (see Section 2.7).

### A2    Water stage observation uncertainty

To illustrate the uncertainty in WSE observations, Figure A7 shows a comparison of the WSE at 100B observed at two nearby locations. The results show that the ME between observation 2 and observation 1 is 3.219 cm, however, the standard deviation

between the two observations is 11.555 cm (Figure A7b). We argue that the large standard deviation is attributed to a small time uncertainty during the observation. This can be proved by Figure A7c which shows that the standard deviation reduces to 4.763 cm if the time history in observation 2 is shifted by 39.3 minutes. However, Figure A7c also means the time shift does not contribute to a large uncertainty in its mean value as the ME is always in the range 3.08 cm – 3.22 cm for any time

shift between -120 minutes and 120 minutes. As the mean value of WSE is used to calibrate roughness, the above results thus demonstrate that the current WSE survey technique does not bring significant uncertainty for roughness quantification but could result in a large difference in standard deviation, mean absolute error, and root mean square when comparing the modeled WSE to observed ones. Actually, if we do an alignment of observation 2, i.e., shifting observation 2 by 39.3 minutes in time and adding 3.219 cm to its value, we see that the difference between observation 1 and such an aligned WSE is clearly reduced ( Figure A7b).

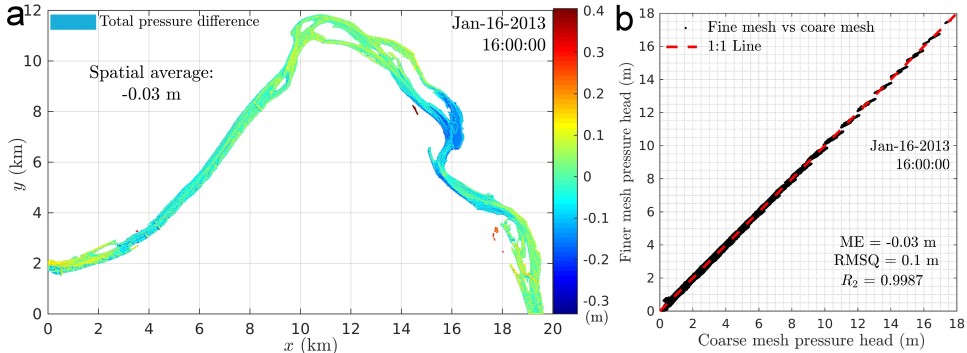

**Figure A1.** Distribution of the difference between total pressure modeled with a fine mesh and a coarse mesh (a), and the 1:1 plot of the total pressure from the fine mesh and the coarse mesh (b).

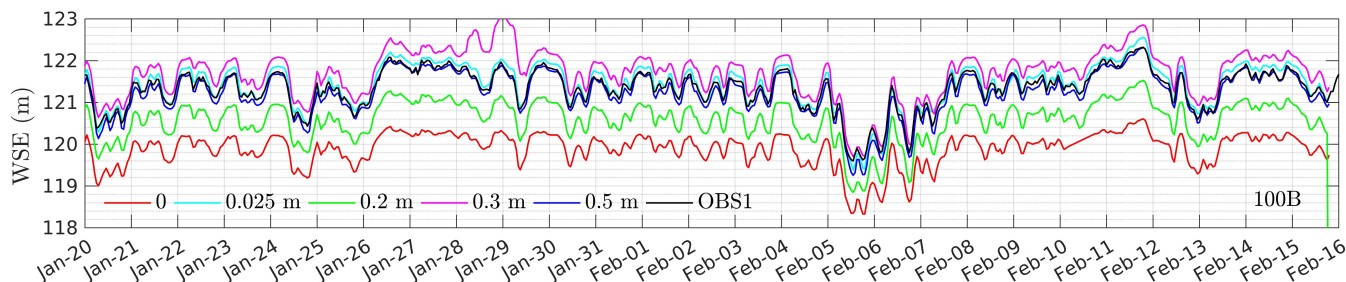

**Figure A2.** A comparison between observed WSE at 100B and modeled ones using different roughness height.

**Table A1.** Horizontal coordinates and bed elevation of survey locations.

| Station | $x$ (m) | $y$ (m) | $z_b$ (m) |
|---|---|---|---|
| 100B | 555.63 | 1619.60 | 117.69 |
| 100N | 6759.03 | 5882.76 | 116.26 |
| 100D | 8516.19 | 8082.07 | 119.05 |
| LI | 12580.24 | 10298.23 | 113.74 |
| 100H | 13260.85 | 9756.13 | 114.45 |
| 100F | 16676.44 | 4429.60 | 110.77 |
| 100HD | 15451.55 | 7581.22 | 112.61 |

**Table A2.** Coefficients of $k - \omega$ turbulence model

| $\beta^*$ | $\alpha_{\omega 1}$ | $\alpha_{\omega 2}$ | $\alpha_{k1}$ | $\alpha_{k2}$ | $\beta_1$ | $\beta_2$ | $\gamma_1$ | $\gamma_1$ | $a_1$ | $b_1$ | $c_1$ | $C_\mu$ |
|---|---|---|---|---|---|---|---|---|---|---|---|---|
| 0.09 | 0.5 | 0.856 | 0.85 | 1 | 0.075 | 0.0828 | 0.555556 | 0.44 | 0.31 | 1 | 10 | 0.09 |

**Table A3.** Roughness parameters used in MASS1/2 and associated model accuracy.

| Survey | MASS1 Calibration | | | | MASS1 Validation | | MASS2 Calibration | | | | MASS2 Validation | |
|---|---|---|---|---|---|---|---|---|---|---|---|---|
| Station | n | $k_s$ | ME | MAE | ME | MAE | n | $k_s$ | ME | MAE | ME | MAE |
| 100B | 0.033 | 13.1 | -0.2 | 17.6 | 24.0 | 26.0 | 0.038 | 30.5 | -3.8 | 11.7 | 7.8 | 12.5 |
| 100N | 0.0313 | 9.5 | 0.0 | 15.6 | 27.0 | 30.0 | 0.035 | 18.6 | -2.8 | 12.8 | 4.5 | 8.4 |
| 100D | 0.034 | 15.6 | 0.2 | 16.1 | 19.0 | 22.0 | 0.034 | 15.6 | -2.7 | 10.2 | 3.3 | 4.7 |
| LI | 0.0346 | 17.3 | 0.1 | 4.8 | NA | NA | 0.027 | 3.9 | -2.2 | 11.8 | 2.5 | 4.1 |
| 100H | 0.0265 | 3.5 | 0.2 | 6.4 | -1.0 | 4.9 | 0.027 | 3.9 | -2.7 | 6.6 | 0.2 | 0.6 |
| 100F | 0.0296 | 6.8 | 0.2 | 7.9 | 19.0 | 22.0 | 0.03 | 7.4 | -0.8 | 3.9 | 1.9 | 3.9 |
| Range | - | - | -0.2 – 0.2 | 4.8 – 17.6 | -1.0 – 27.0 | 4.9 – 30.0 | - | - | -3.8 – -0.8 | 3.9 – 12.8 | 0.2 – 7.8 | 0.6 – 12.5 |

Units for $n$, $k_s$, ME, and MAE are s/m$^{1/3}$, cm, cm, and cm, respectively. The time periods for MASS1 calibration and validation are 10/3/2010 – 3/7/2011 and 7/1/2011 – 9/1/2011; and those for MASS2 are 10/4/2010 – 10/10/2010 and 1/4/2011 – 1/7/2011. Values of $n$, ME, and MAE can be found in Ref. Niehus et al. (2014). Values of $k_s$ are used as a reference and calculated by $n = \frac{1.219}{a\sqrt{g}} k_s^{1/6}$ with a = 8.4 as discussed in Section 4.1.3.

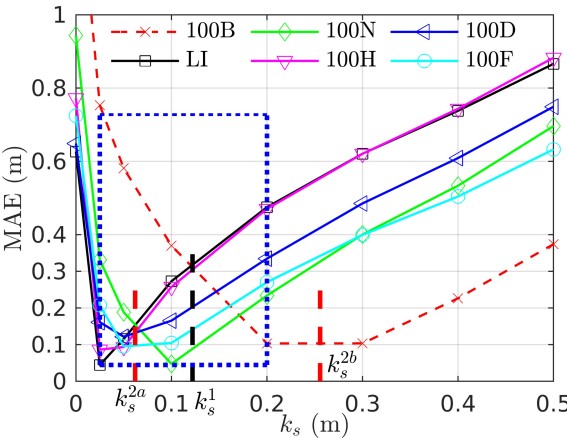

**Figure A3.** The variation of mean absolute error (MAE) between modeled and observed WSE at six locations using different roughness parameters. Black and red vertical lines represent the optimal roughness height using one-ks and two-ks strategy.

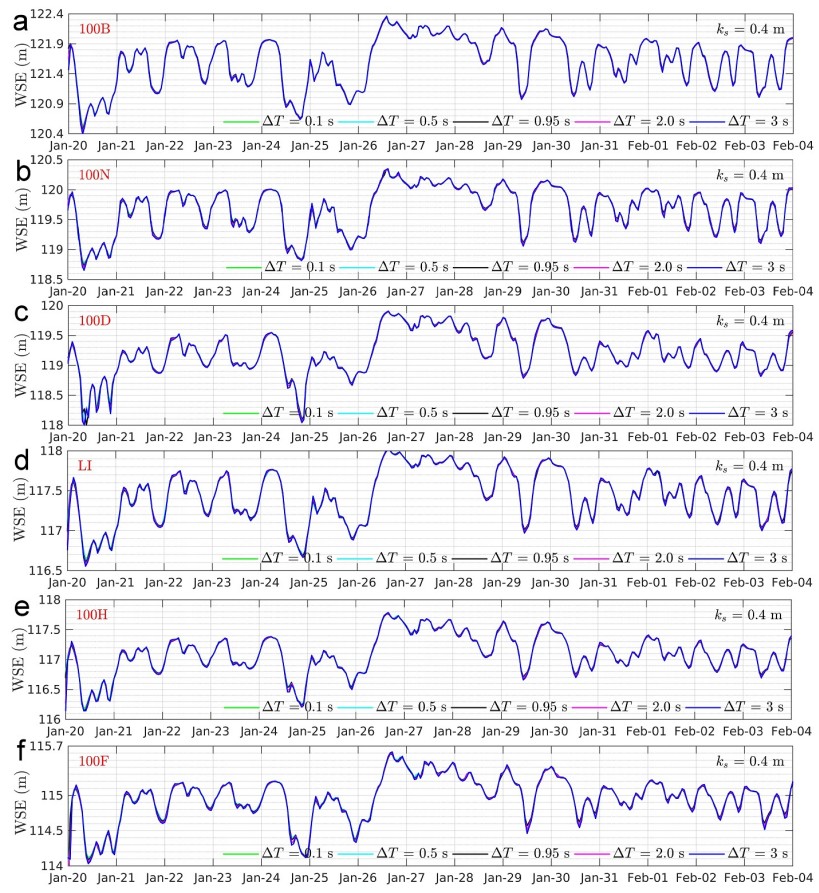

**Figure A4.** A comparison of WSE at different time step at 100B, 100N, 100D, LI, 100H, and 100F.

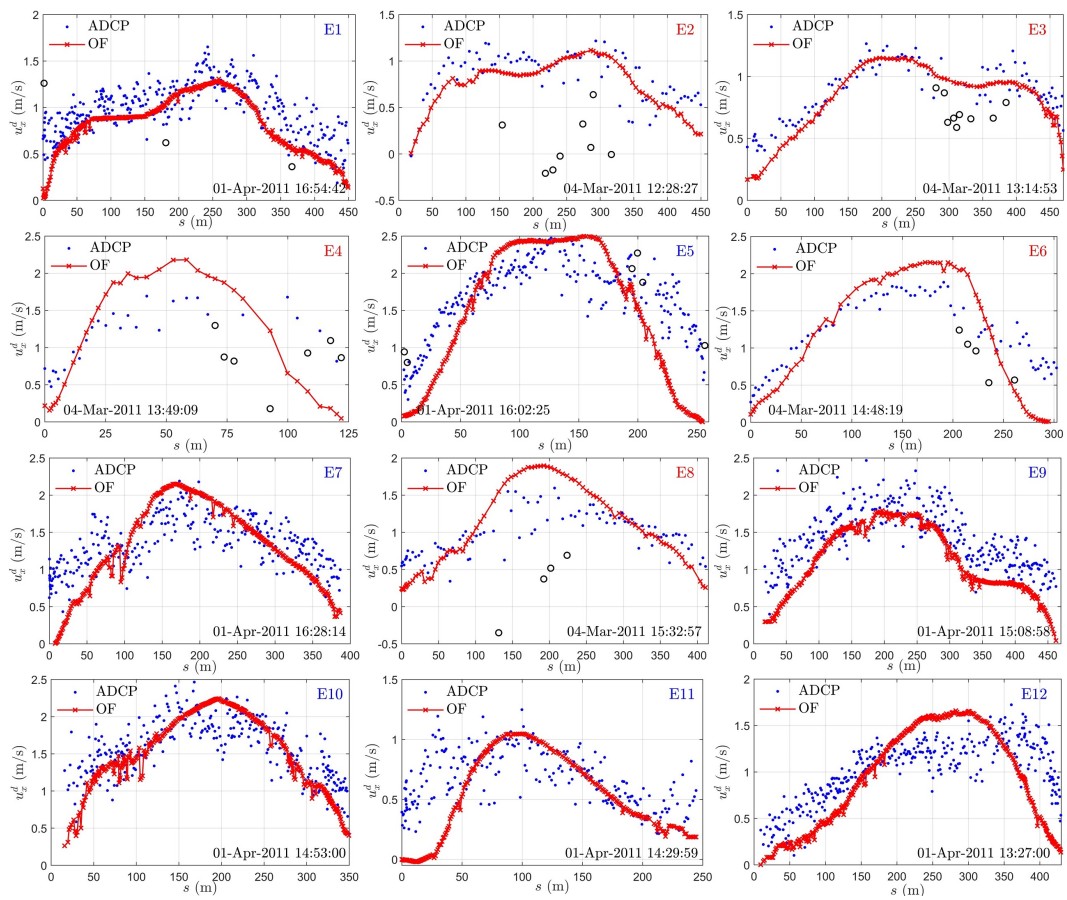

**Figure A5.** A comparison of depth-averaged velocity component along $x$ from ADCP surveys and CFD modeling at E1 - E12. Black circles denote measured outliers visually determined from Figure A5 or A6.

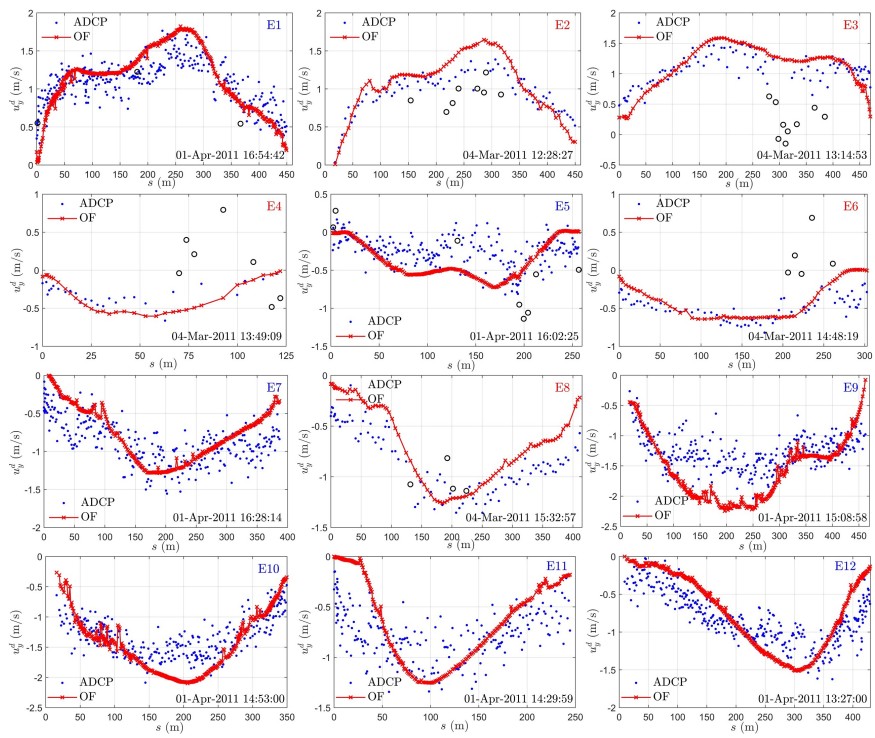

**Figure A6.** A comparison of depth-averaged velocity component along $y$ from ADCP surveys and CFD modeling at E1 - E12. Black circles denote measured outliers visually determined from Figure A5 or A6.

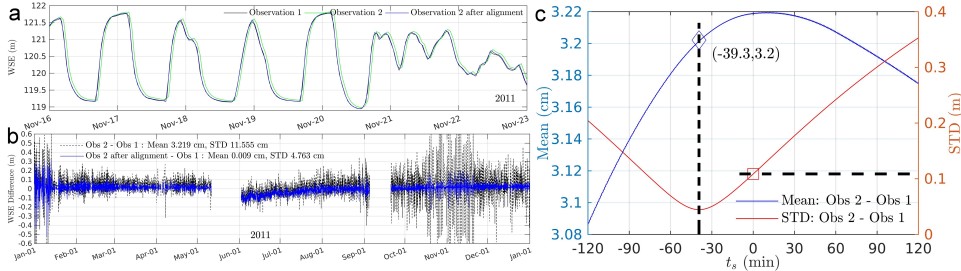

**Figure A7.** A comparison of WSE at 100B from observation 1, observation 2, and observation 2 after alignment (a), the differences in WSE between observation 1 and observation 2 and that between observation 1 and observation 2 after alignment (b), and the mean and standard deviation between observation 1 and observation 2 with a time shift $t_s$ (c).

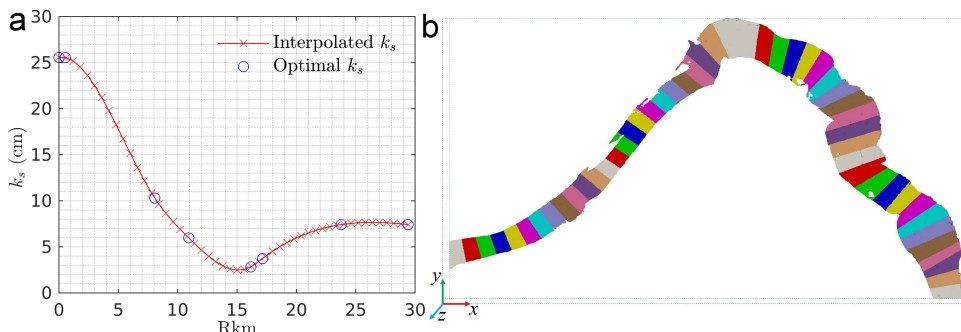

**Figure A8.** The roughness height on 50 pieces of stream interpolated from the 6 globally optimal roughness parameter (blue circle) (a) and the decomposition of the streambed into 50 pieces (b).

*Code and data availability.* OpenFOAM setups, data, and Matlab code are available at ESS-DIVE data archive https://doi.org/10.15485/1819956.

*Author contributions.* Y. C.: Conceptualization, methodology, software, validation, analyses, investigation, writing; J. B.: conceptualization, analyses, discussion, supervision, project administration, manuscript review; Y. F.: discussion, manuscript review; W. A.: discussion, 1D model data generation; H. R.: discussion, 1D model data generation, water stage data preparation; X. S.: discussion; Z. D.: geometry data preparation; Z. H.: discussion; X. H.: discussion; T. S.: project administration, funding acquisition, discussion, manuscript review.

*Competing interests.* The authors declare that they have no known competing financial interests or personal relationships that could have appeared to influence the work reported in this paper.

*Acknowledgements.* This research was supported by the United States Department of Energy (DOE) Office of Biological and Environmental Research (BER), Subsurface Biogeochemical Research program, through the PNNL Subsurface Science Scientific Focus Area (SFA) project (http://sbrsfa.pnnl.gov/). PNNL is operated for the DOE by Battelle Memorial Institute under Contract No. DE-AC05-76RL01830. Data and
modeling products are retained in the SFA data management system and are available from the authors on request. The high-performance computation resources were provided by EMSL Cascade.

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
