# Peer review of "Modeling of streamflow in a 30-kilometer-long reach spanning 5 years using OpenFOAM 5.x"

_Geoscientific Model Development, 2021_

## Author Comment (AC2)

[revised manuscript text omitted]
} e_x + \frac{\partial}{\partial y} e_y + \frac{\partial}{\partial z} e_z$  represents a spatial operator with  $e_x$ ,  $e_y$ , and  $e_z$  denoting unit vectors along x, y, and z directions. Also denoted are time average flow velocity (u), surface tension coefficient ( $\sigma$ ), interface curvature ( $\kappa_{\alpha}$ ), gravity acceleration (g), spatial coordinate (x), dynamic pressure ( $p_d$ ), and dynamic turbulent viscosity ( $\mu_t$ ). Specifically, the interface curvature is calculated by  $\kappa_{\alpha} = -\nabla \cdot (\frac{\nabla \alpha}{|\nabla \alpha|})$ , the dynamic pressure  $p_d$  is defined as  $p_d = p - \rho g \cdot x$  with p denoting the total pressure,  $u_r$  is an artificial velocity whose definition can be found in Deshpande et al. (2012). The dynamic turbulent viscosity is determined by the  $k - \omega$  shear stress transport (SST) model (Menter et al., 2003; Wilcox, 2006; CFDDirect, 2017).

**2.3 Mesh generation and quality control**

130

125

A good mesh quality is a crucial factor controlling computational stability and efficiency, especially for free surface tracking o in large-scale river modeling over a long period (Deshpande et al., 2012). In this work, the mesh is generated using a twostep generation strategy, which first generates a structured background mesh and then removes all cells totally outside a given geometry (a river bathymetry in our case). Specifically, the background mesh is generated with a horizontal mesh resolution as 20 m along x and y. Such a resolution is identical to the horizontal resolution of the LiDAR-measured digital elevation model (DEM). The vertical mesh resolution is set as  $\Delta z = 1$  m by balancing modeling accuracy and computational costs. One

135

model (DEM). The vertical mesh resolution is set as  $\Delta z = 1$  m by balancing modeling accuracy and computational costs. One extra mesh resolution, 20 m× 20 m× 0.5 m, is also created to investigate the sensitivity of modeled riverbed pressure to mesh resolution (see uncertainty analyses in Appendix A1 and Figure A1). Figure 2 shows the horizontal and vertical mesh in the computational domain. It is observed that the aspect ratio for horizontal (x and y) grid sizes is 1, but in the vertical direction it is 20. Figure 2c also shows that the zig-zag grid does not overlap with the riverbed, whose effect on flow is further discussed in the roughness calibration (see Section 2.4).

---

## Author Response (AR1)

Dear Editor,

We thank your efforts to contact the reviewers to provide many useful comments to improve our paper. The comments from the two reviewers have been incorporated into the paper. Here we provide a point-by-point response for all comments from the reviewers as follows. We also provide two versions of the revised paper for your reference. One is the track-change version and another one is the final clean version. Both files are generated using the Copernicus Latex template and could be easily regenerated from the attached .tex file. In the track change version, texts highlighted with "a" (red), "r" (blue),"d" (gray) represent adding new texts, replacing existing texts, and deleting texts, respectively.

Thanks for your efforts. Please let me know if you need more materials from me.

Yunxiang.

**[Reviwer 1: Comments, responses, and changes]**

**[Reviewer 1 comments 1]:** The work in this paper uses relevant field data to provide appropriate calibrations and constraints in the development of a 3-D CFD model of the streamflow in a reach at large length and time scales. I applaud this 'grand challenge' effort to establish a significant benchmark for the accurate long time/length scale prediction of streamflow.

**[Response]:** We thank Prof. Vaughan Voller's recognition of the importance of this work. Yes, this work aims to demonstrate that river hydrodynamics at tens-kilometer and multiple-year scales can be accurately predicted by combining observation data (bathymetry, water stage, and discharge), open-source CFD code (OpenFOAM), carefully designed roughness calibration strategies, and high-performance computing (HPC). Additionally, we provide reproducible procedures and guidelines regarding data acquisition, mesh generation, distributed roughness calibration, and parallel computation leveraging commonly accessible survey techniques (e.g., LiDAR, water level logger, ADCP) and computing resources. These procedures, guidelines, techniques, and computing resources altogether provide a generic and repeatable framework for modeling the streamflow at spatiotemporal scales that are essential for both local-scale water infrastructure design, construction, and operation and evaluating the impacts of climate- and human-induced discharge variations on local river hydrodynamics.

**[Reviewer 1 comments 2]:** I have a number of comments for the authors to consider. In my view, unless the authors so wish, none of these require changes to the current manuscript.

**[Response]:** We thank the reviewer's invaluable suggestion. The references have been incorporated into the Introduction section in the revised version of the paper. Please check the red text between lines 64-69 in the Track-changes version.

**[Reviewer 1 comments 3]:** The authors provide a comprehensive survey of the literature of the full range of previous modeling efforts for streamflow. One set of work which may have been overlooked is a series of papers by Kang, Sotiropoulos and others.

Kang, S., A. Lightbody, C. Hill, and F. Sotiropoulos (2011), High-resolution numerical simulation of turbulence in natural waterways, Adv. Water Resour., 34(1), 98– 113.

Kang, S., and F.Sotiropoulos (2011), Flow phenomena and mechanisms in a field-scale experimental meandering channel with a pool-riffle sequence: Insights gained via numerical simulation, J. Geophys. Res., 116, F03011, doi:10.1029/2010JF001814.

Kang, S., and F.Sotiropoulos (2011), Assessing the predictive capabilities of isotropic, eddy viscosity Reynolds-averaged turbulence models in a natural-like meandering channel, Water Resources Research, 48, https://doi.org/10.1029/2011WR011375

(full disclosure I was on Kang's PhD committee)

These works compare 3-D free surface/ turbulence (both RANS and LES) model predictions of streamflow with highly resolved measurements in a natural channel. The scale of these calculations is smaller than the current work (10 's m as opposed to 10's km) but the conclusion of the Kang et al work does point out possible accuracy issues in using time averaged turbulence models.

Some questions in this regard, Is a RANS model sufficient for the task at hand? Would LES improve predictive performance? Is LES currently feasible at the scale of the current simulation?

**[Responses]:** We thank the reviewer's insightful questions. Here are our thoughts on these questions.

From Kang et al's AWR and WRR work, they showed that the streamwise/transverse velocities and turbulent kinetic energy profiles are reasonably predicted using LES and SST-based RANS. The main differences between LES and RANS-SST lie in predicting the 3D secondary flows. They concluded that this discrepancy depends on the level of turbulence anisotropy that is further affected by streambed geometry (roughness and channel deepening) and flow rate. In particular, in the riffle regions, the 3D flow is dominated by the large-scale rough elements that generate large anisotropic turbulence and vortices; in the pool regions, the RANS can predict the curvature-induced secondary flow, but cannot predict the other features that are caused by the anisotropic turbulence induced by channel deepening at a bank full (with depth 0.3 m) flow rate. However, such a difference is reduced at a low flow (with depth 0.1 m) as shown in the ADW paper. From these results, it can be concluded that velocities and turbulence statistics can be reasonably predicted using both RANS-SST and LES, though the complex secondary flows induced by small-scale streambed sediments and local topography variations (e.g., channel deepening) are not well predicted by RANS. For the task reported in this paper, we argue that the RANS model is sufficient for predicting water surface elevation (or depth) and velocities, whose accuracy is supported by Figs. 5,6,8,9,10. Though Kang et al showed the 3D flow structures may not be predicted by RANS (at bank full), the comparison of

velocity distributions at the 12 ADCP survey locations (Fig.7) indicates that no significant differences are observed between the model and measurements. Therefore, RANS with SST is likely sufficient for the current work.

Regarding if LES improves the performance, I would argue that LES may not improve the model performance due to (a) the limited accuracy in streambed topography surveys and (b) the uncertainties in velocity measurements. The three papers by Kang et al. demonstrated that the prediction of complex secondary flows can be improved if the small-scale (cm-scale) streambed rough elements can be accurately measured in topography surveys and their effects on the turbulence can be resolved by LES. Obtaining the details of rough elements in the field-scale lab facility with 0.1 to 0.3 m depth (as shown in Kang's work) is feasible, however, it is not practical to measure rough elements at km-scale natural rivers with water depth range 0-20 m (the river section studied in this work). In the current work, the river bathymetry is measured using a LiDAR whose spatial resolution cannot capture the sediment-scale (mm to dm) rough elements. Due to the missing information of small-scale rough elements, their effects on the flow must be calibrated by matching the model predicted water stage with observed ones. Replacing the RANS with LES cannot avoid such a calibration procedure. The ultimate model accuracy, therefore, depends on the accuracy of water stage observations and the roughness wall model (need to use rough wall model to account for the missing roughness effect), but not the scales resolved by LES. Additionally, the evaluation of a model's performance also depends on the accuracy of the measurement data. This is especially true for validating the model with field velocity surveys because typical velocity survey techniques, e.g., boat towed ADCP, usually have large uncertainties. It is difficult to know whether the model performance improves or not without improving the velocity measurement accuracy. In Le, T. B. et al (Large-eddy simulation of the Mississippi River under the base-flow condition: hydrodynamics of a natural diffluence-confluence region, JHR, 2019), they simulated the flow in a 3.2 km long reach to a quasi-steady state using LES with a mesh resolution 0.5 m, 2.37 m, and 0.094 m along x, y, z directions. Though the vertical mesh resolution is sufficient to resolve the effects of small-scale sediments (let's assume the sediment details are captured by the topography survey) on the flow, the velocity prediction accuracy is not improved compared to using a much coarser mesh with RANS in our paper (please compare Fig 12 in Le, T. B. et al with Fig. 8 in our paper). Please refer to lines 305-311 for the details of this discussion as well as Le, T.B. et al's paper.

To my best knowledge, the work by Le, T. B. is likely the largest spatial scale (3.2 km) where LES has been applied to. It is possible to extend such a scale to the scale (30 km) studied in the current work but will be likely not feasible to extend it to a multiple-year scale because of the limitation of the small time step required by LES. Modeling natural rivers using LES also needs to address the dynamically changing water surface that necessitates efficiently integrating LES with VOF. Additionally, LES likely improves the model's accuracy only if the details of streambed topography can be accurately measured. In recent years, structure-from-motion photogrammetry has demonstrated its capability in obtaining mm- to cm-scale resolution streambed at dry and shallow flow conditions up to 40 km reach. Integration LES with SfM-derived topography will likely be a key step to fully utilize the power of LES (in capturing the anisotropic turbulence induced by irregular strembed sediments). Discussions on this can be found in Section 4.1.5.

**[Reviewer 1 comments 4]:** The authors provide a nice explanation of how they balanced the modeling efforts between computational efficiency and predictive accuracy. Much of this focuses on how the code was constructed to segment the calculations and reduce the CPU requirement. Of course, in a large modeling study, of the scale reported here, the actual CPU time may only be a part of the overall effort used. Could the authors comment on the resources to set up the model (meshing, calibration etc) and the resources required to validate the model ; Were these of the same order as the CPU?

**[Responses]:** We thank the reviewer raise these questions. In general, we divided the overall time costs into model setup time, simulation time, and post-processing time.

In the model setup step, we have some key tasks including (a1) converting DEM data (in ASC format) to a triangulated surface format (STL); (a2) setting up a script for the snappyHexMesh in OpenFOAM to generate a mesh based on the input STL; (a3) developing a Matlab code to extract the water depth and velocity data from a 1D hydrodynamics model (Mass1), (a4) writing these data as time-variable boundary conditions for velocity and volume fraction at the inlet and outlet for OpenFOAM; (a5) developing a Matlab code to read the OpenFOAM mesh, split the streambed boundary faces into N regions (see details between lines 412-440) based on the input water level logger location coordinates, and assign locally or globally optimal roughness values for each region through a rough wall model; (a6) copying the final globally optimal roughness value case for all months between 2013-2015 (36 months) and 2018-2019 (22 months), and setting up for the starting and ending time for each simulation in OpenFOAM; (a7) submitting all cases for parallel simulations.

Though most of the steps take little time, step a5 takes considerable time because we need to implement the local roughness optimization and adjustment strategy in this step (see Section 2.4). In the local roughness optimization step, we firstly run 8 simulations for the time period between Jan-20 to Feb-16, 2011 by assigning 8 different uniform roughness values (0, 0.025 m, 0.05 m, 0.1 m, 0.2 m, 0.3 m, 0.4 m, and 0.5 m) for the whole streambed. Then we determined the locally optimal values for each observation location based on Fig. 3b., which finally results in the case OF0 as shown in Table 1. In the roughness adjustment step, 5 additional simulations are conducted based on the case OF0 to adjust the roughness values at upstream locations as shown in cases OF1-OF5 in Table 1. The roughness values derived via this step are used in step a6 for parallel simulations. In total, the 8 simulations conducted in the roughness optimization step consumed 254,000 CPU hours with each case consuming around 32,000 CPU hours. The 7 simulations implemented in the roughness adjustment step used 52,000 CPU hours with each case consuming 7,400 CPU hours. The cases in the adjustment step spend around 4.3 times less time compared to those used in the roughness optimization step because larger time step (3s vs 0.95 s) and improved linear solvers are used in the adjustment step. With the optimal distributed roughness values, we run the Matlab code developed in step a4 to generate 58 OpenFOAM cases for all months during 2013-2015 and 2018-2019 for parallel simulations. The total time required in this step is around 6 days (or 1.1 million CPU hours) as reported in Section 4.4. Combining the time used in the calibration step, the overall computing cost is around 1.4 million CPU hours with around 21% used for model setups and calibration. The

ratio of model calibration time to the model running time is around 28%. This information has been added into the paper at line 609.

When all simulations are completed, it takes extra time to post-process the results from OpenFOAM. The time consumed in post-processing depends on the dimension of the data we need to use. For the water stage data reported in Figs. 5,6,9,10, they are interpolated from the OpenFOAM results at 6 locations, which takes a few hours to obtain the data. For the velocity data reported in Figs. 7-8, the OpenFOAM results are interpolated to 12 cross-sections (thousands of coordinates) at two specific hours. This post-processing takes less than 1 hour using a Matlab code developed for this purpose. For the pressure data reported in Fig. 11, they are interpolated from the OpenFOAM results to a 2D uniform grid with 1.2 million points using a Matlab code. It takes around 8-12 hours to extract all pressure data during 2013-2019.

In our work, we have developed a series of Matlab functions and scripts to automate the model setups, parallel job submissions, and post-processing. The actual time required for the simulations and post-processing depends on the spatiotemporal scales to be investigated. But for the model setups and calibration, it only takes 2-3 days to finalize the case setups for final simulations by running our scripts step by step. This conclusion has been demonstrated in another work where we apply the same framework to another 30 km-long reach in the Columbia River.

To apply the framework for other rivers, the following resources are necessary: (1) digital elevation model of the target river; (2) water stage surveys at a few locations along the target river; (3) velocity measurements at a few cross-sections along the river; (4) velocity and depth information at the inlet and outlet locations for the target river (a 1D hydrologic model was used to extract this information in this work); (5) computing resources. For practical applications, it is better to first check with national or local water agencies or open data repositories regarding the availability of these resources.

**[Reviewer 1 comments 5]:** The authors close by correctly pointing out the possible benefit of their approach in assessing impacts of climate change. They could be a little more specific here. In particular, are the space and time domains presented here sufficient for meaningful climate change scenario modeling? If not, what time scales and reach sizes would be meaningful?

**[Response]:** We thank the reviewer's insightful comments. A natural river system is strongly affected by the discharge that is further controlled by the overall water balance, human usage, and climate forcing such as precipitation. Existing global hydrological models such as the Variable Infiltration Capacity (VIC) model have enabled predicting global stream discharge at a spatial scale of 5-10 km and a time scale of 3 hours to one day spanning 40 years with precipitation as the main climate forcing (Yang, Y. et al., Global Reach-Level 3-Hourly River Flood Reanalysis (1980–2019, BAMS, 2021). The impact of human activities on global water resources has also been evaluated by integrating the VIC model with anthropogenic modules, though at coarser (55 km) resolution (Droppers, B., et al., Simulating human impacts on global water resources using VIC-5, GMD, 2020). With the discharge output from these models, it is straightforward to evaluate the impacts of climate and human activities on river hydrodynamics by constraining the river CFD model with the discharge from global

hydrologic models. In the present work, the space and time domains are 30 km and around 9 years (2011-2019) with a spatial resolution of 20 m and a temporal resolution of 3 seconds. This information means that the space and time scale studied in this work fall within the scales (global 5 km and 3-hour resolution discharge data available according to Yang, Y. et al) of the latest global hydrological modeling. Therefore, the current CFD framework can be meaningfully linked to global hydrologic models to understand their impacts on river hydrodynamics and associated biogeochemical processes. We have incorporated this message into the revised version of the paper (see red texts between 635-640).

**[Reviewer 1 comments 6]:** While outside of the scope of the current paper. In future work it might be worthwhile to compare the performances (CPU/predictions) of the proposed 3D RANS/fee surface calculations with the more widely used 1-D and 2-D streamflow codes noted in the literature review. Vaughan Voller, University of Minnesota.

**[Response]:** We thank the reviewer's suggestion for future work. To keep the paper short, this paper only compared the water stage prediction from the proposed 3D model with another 1D (Mass1) and 2D (Mass2) model. But it will be of great practical value to compare our model with other widely used 1D/2D models in terms of their long-term performance and computational costs.

**[Reviwer 2: Comments, responses, and changes]**

**[Reviewer 2 comments 0]:** This manuscript presents a detailed implementation of a 3D CFD model for a 30-km transect of the Columbia River. The model is calibrated with surface water elevations (SWE) and validated with SWE and velocity profiles. As part of this effort, an analysis of the model performance at three different time scales was performed: (1) short-term (2012), (2) middle-term (2013-2016), and (3) long-term (2018-2019).

The authors present a novel and practical approach to impose boundary conditions and calibrate roughness scales and shear stresses in large-scale 3D CFD models. After calibration and validation, the authors compared the new model results with previous modeling efforts in the study site. This analysis highlighted the importance of a detailed representation of 3D processes in channel flow modeling. Finally, the authors assessed the relative importance of dynamic and hydrostatic pressure along the reach, focusing on the potential implications for environmental and ecological functions in river systems.

Summary: Overall, this manuscript is well-written, clear, and represents an important contribution to the field of river hydraulics. The implementation approach is well-documented and reproducible. This will make for a significant contribution to GMD.

I include a commented pdf with editorial suggestions. In the following, I present two significant comments that require attention and a series of general observations.

**[Responses]:** We thank the review's comments and detailed suggestions in the attached pdf file. They are very helpful to improve the paper.

**[Reviewer 2 major comments 1]:** The authors' argument for using a spatially-variable roughness (ks) is that this parameter is expected to vary for a complex river system with heterogeneous bedforms and high curvature. I agree with this argument. However, the selection of roughness regions seems guided by geometric convenience and data availability and not the spatial variations that are reasonable controls for ks. For instance, I would expect that a characterization of the different depositional environments and corresponding bedforms would provide a better guide for selecting roughness zones. The PNNL team has previously performed such classification for the study reach, which could be used. The main issue here is that given the complexity of this model, the authors are likely dealing with a non-unique solution, and the computational burden prevents them from analyzing this in detail. Some discussion in this regard would be valuable in this manuscript and essential to guide future model refinements. I suggest including some discussion in lines 415-422.

On a related note, I wonder how sensitive is the spatial variation of stage and velocity to the calibrated ks within a roughness region. Again, the authors assess this sensitivity at the point scale, where the SWE observations are available, but the overall response may be insignificant.

**[Responses]:** We thank the reviewer's invaluable comments and suggestions. The reviewer is correct that we designed the model calibration procedure based on geometric convenience and data availability. Though it will be of great value to measure water stage at diverse locations (e.g., shorelines and thalwegs) and environments (e.g., gravels, dunes, mixed gravel and sand, vegetations), logistics and safety issues need to be considered in field surveys. For example, to install the water stage survey devices safely, we cannot walk to the depth water (>2 m) regions. Figure 1c shows the exact locations of each stage survey locations (red stars), and we can see that all these locations are very close (<100 m) to the river banks. Even though they are very close to the shorelines, the water depth at these locations could vary between 1 m – 4 m most of the time which can be observed in Figure 5c. Furthermore, devices deployed too close to the river centerline also impose dangers to normal human activities (e.g., boating, shipping, kayaking, etc.) and could be more easily destroyed by floating objects (woods) that more likely occur in the river center regions. Due to these reasons, the water stage deployment was guided by geometric convenience in this work. In addition, the water stage surveys were mostly conducted 10 years ago, therefore, the strategy of how to divide the riverbed into multiple roughness subregions is designed based on existing survey locations, but not guided by the characteristic river planform (sinuosity/curvature/narrow-width variations) and depositional environments. For future applications, however, it is recommended to first design the survey locations based on characteristic river planform and depositional environments (if logistics and safety are not an issue), and then apply the modeling framework presented in this paper. Relevant discussions on these issues have been added to lines 450-456 in the attached revised version of the paper.

Regarding the non-unique solution issue, a discussion on how to address this issue is added in lines 446-456. The key takeaway message is that the final model performance is controlled by both local

optimal roughness estimation and local roughness adjustment. The first step is purely based on the error diagram and it will provide a unique roughness value for each location. The second step needs to adjust the roughness estimated from the first step based on (1) our physical understanding of how the characteristic river planform and depositional environments affect the water stage and (2) how much model more error at each location. As this is a case-by-case and trial-and-error procedure, it may not guarantee a unique solution. However, with a better design of the survey location distribution, it is likely to achieve a high model accuracy even without the second step. The higher the accuracy in the first step, the less important the second step. However, if the second step is a must, the latest machine learning approaches such as parameter learning and reinforcement learning could be potential directions to explore in the future.

To clarify the sensitivity of stage and velocity variations to roughness, a new section 4.1.3 and Figure 12 are added. Based on the discussions, we have the following answers: (a) the spatial variation of water stage is not sensitive roughness at locations near the river valleys, however, it is sensitive to the roughness at locations near the river banks; (b) the absolute value fo the water stage near the river valley regions vary 18% to 28% in response to changing roughness from 0 to 0.5 m, however, the water stage near the river banks experiences significantly more variations; (c) the spatial distribution of free surface velocity magnitude is not sensitive to non-zero (zero roughness is a toally different story) roughness, however, its maximum value could vary 25% to 30% when the roughness height increases from 0.025 m to 0.5 m. In short, the water stage near the banks and the velocity near the river valleys are very sensitive to roughness. By contrast, the water stage near the valley and the velocity near the banks are less sensitive to roughness. More details of these discussions can be found in lines 458-490.

**[Reviewer 2 major comments 2]:** The idea of analyzing the relative importance of dynamic and hydrostatic pressure is an excellent illustration of the importance of these models to gain a mechanistic understanding of exchange processes along the sediment-water interface. I commend the authors for including this analysis! However, I expect the conclusions regarding the exchange to be incorrect. The reason for my skepticism is that the exchange process is driven by gradients in the pressure distribution and not by its magnitude. In other words, I suggest that the authors revise this analysis and focus on the spatial variability of the pressure gradients.

**[Responses]:** We thank the reviewer's insightful questions. The reviewer is correct to point out that it is the pressure gradient at the riverbed that drives the exchange fluxes. The question is how to obtain the pressure gradient. Direct measurements of the pressure gradient at natural streambeds are challenging, therefore, numerical models are usually used to simulate the pressure gradients. The pressure (and gradient) at the riverbed is the summation of hydrostatic and hydrodynamic pressure. Depending on how to deal with the dynamic pressure, existing models that study surface-subsurface interactions can be categorized into three types: no dynamic pressure models, one-way coupled dynamic pressure models, and two-way coupled dynamic pressure models. The first type of model ignores the dynamic pressure. The second type of model solves the 3D hydrodynamics equations for the surface water and then uses the computed total pressure (the summation of hydrostatic pressure and dynamic pressure) as a boundary condition to drive the subsurface flow without

considering the feedback of flow from the subsurface to the surface. The third type of model is similar to the second type but allows the flow from the subsurface to the surface. Though the third type of model can provide the pressure gradients at the riverbed, they are usually very computationally expensive and numerically unstable, and thus are currently only used at laboratory-scale and idealized flow conditions. A more commonly used approach is the one-way coupled model. In this approach, it first assumes the dynamic pressure gradient at the streambed is zero, and then solves the Navier-Stokes equation to obtain the magnitude of the dynamic pressure. In this paper, we also set a zero dynamic pressure gradient at the riverbed (see line 213). With the dynamic pressure at the riverbed, the one-way coupled model solves the subsurface flows (e.g., 3D Richards equations) by setting the total pressure at the riverbed as a boundary condition. This finally provides the pressure gradients (due to both hydrostatic and dynamic pressure) at the streambeds. Therefore, it is important to evaluate the relative importance of dynamic pressure to static pressure to understand how dynamic pressure affects the exchange fluxes. Actually, in another work, we have used the total pressure computed from the CFD model to drive a subsurface flow model and reported the impacts of dynamic pressure on exchange fluxes. To address the reviewer's concerns, we have added the above discussions in lines 555-581. The effects of dynamic pressure on exchange fluxes can also be found in Bao, J. et al., Modeling framework for evaluating the impacts of hydrodynamic pressure on hydrologic exchange fluxes and residence time for a large-scale river section over a long-term period. Environmental Modelling & Software, 2022, 148,105277.

[Reviewer 2 general comments]:

[Reviewer 2 comment 3]: p7, l 180: For clarity, the roughness elements directly resolved are larger than 1m in "the vertical direction." To be precise, you could include "the vertical direction" in the text since the horizontal resolution is much lower (20m).

[Response]: Yes, we have corrected this sentence at Line 185 in the revised version of the paper.

[Reviewer 2 comment 4]: p9, l 206: the velocity components for the outlet seem to have the x- and y-direction components mixed

[Response]: Velocity is a three-component vector with the first, second, and third components denoting the velocity component along x, y, and z-direction. At the outlet, the velocity along x (east) and z (towards water surface) directions are zero, while the velocity along y-direction (north) is non-zero. Therefore, the velocity vector is (0, -uout, 0) and the equation at line 210 is correct.

[Reviewer 2 comment 5]: p 11, l 255: To better illustrate the potential presence of systematic biases, I suggest plotting error vs. stage. For example, the bias for low SWE in Figure 9 will be more evident with this metric.

[Response]: To illustrate if using a water stage causes misunderstanding of the model performance (visual comparison, biase, mean error, mean absolute error, etc.), we add new subfigures in Figure 5 and Figure 9 by showing the comparison of water depth from the model and observations as well

as their 1:1 plots. Both Figure 5d and 9d indicate that the model does not have systematic biases no matter using water stage or depth as an evaluation metric (see lines 266-268 and 349-351).

**[Reviewer 2 comment 6]:** p 11, l 261 (and throughout the text): Using the tilde symbol "~" for value ranges is somewhat unconventional. I suggest using a dash "-"

**[Response]:** We thank the reviewer's suggestion. Yes, we replaced all "~" with a middle-sized dash to better represent the ranges but avoid confusing it with the minus symbol.

**[Reviewer 2 comment 7]:** The authors use SWE to assess the model performance; however, this metric could be misleading, and I wonder if the water depth is a better alternative. In particular, when calibrating the roughness values, I expect the relative error in water depth to be a more reasonable measure of model performance.

**[Response]:** The response is similar to the comment in Line 255 (comment 5). Based on the new Figures 5c-d and 9c-d, we believe that using WSE and depth conveys an identical message in terms of the model's qualitative and quantitative performance. In addition, we provide the RME, RMAE, and RRMS in Table 2 to quantify the ratio of ME, MAE, and RMS to the average water depth at each location. These quantities are the same when using WSE and depth because the difference between WSE and depth does not contribute to the calculation of ME/MAE/RRMS and RME/RMAE/RRMS.

**[Reviewer 2 comment 8]:** Labels and text in Figure 7 are hard to see.

**[Response]:** Yes, we have revised the figure to make it larger for visualization.

**[Reviewer 2 comment 9]:** p 14, l 290: is it possible that the disagreement for high curvature results from using a constant roughness value for a region with varying depositional characteristics and bedforms?

**[Responses]:** We observed From Figure 12b,e that the non-zero roughness heights do not significantly affect the distribution of the velocity but mainly affect its maximum value. Figure 8 shows that the velocity distribution at E4/E11 is very different from the observed distribution. This disagreement is likely not caused by using a constant roughness. Instead, the poor performance at E4/E11 is likely attributed to the narrow width of the channels at E4/E11. As we use a uniform grid everywhere, mesh resolution may be too small at locations E4/E11. The small width of the channel may also affect the accuracy of the topography survey and boat-towed ADCP surveys, which eventually affects the accuracy of the comparison. Further studies on these issues may be necessary for the future.

**[Reviewer 2 other comments]:** Other comments not mentioned above but included in the pdf document are also addressed. These changes include:

(1) using larger fonts in Fig.1b,c. Plot exact locations of water stage survey on c. Captions are also revised to contain this information. See changes in Figure 1.

(2)  Line 127: add the definition of the artificial velocity ur.
(3)  Line 161: add "turbulent flow" to the "rough regime".
(4)  Line 172: replace "log" by "logarithmic".
(5)  Captions of Figure 5.
(6)  Line 293: using more accurate description of subfigure in Figure 7.

---

## Author Response (AR2)

We thank the editor's comments. We made the following changes based on the editor's comments.

Replacing 'valley' with "thalweg" at lines 464, 467, 471, and 489;

Change "decreasing" to "decrease" at line 477;

Change "no dynamic pressure model" to "models without dynamic pressure" at line 556;

Change "our another work" to "another work of ours" 579.

We also follow the guideline of avoiding parallel usage of red and green in the figures. All figures have been regenerated to replace "green" with yellow or other colors. Non-green colors are not changed. Texts related to green colors are also changed to relate it the new color names.